# Identifying and Solving Conditional Image Leakage in Image-to-Video Diffusion Model

**Min Zhao**[1,3] *, **Hongzhou Zhu**[1,3] *, **Chendong Xiang**[1,3], **Kaiwen Zheng**[1,3],
**Chongxuan Li**[2] †, **Jun Zhu**[1,3,4] †
[1]Dept. of Comp. Sci. & Tech., BNRist Center, THU-Bosch ML Center, Tsinghua University
[2] Gaoling School of Artificial Intelligence, Renmin University of China, Beijing, China
Beijing Key Laboratory of Big Data Management and Analysis Methods, Beijing, China
[3]ShengShu, Beijing, China; [4]Pazhou Laboratory (Huangpu), Guangzhou, China
{gracezhao1997,xiangxyaw,zkwthu}@gmail.com; zhuhz22@mails.tsinghua.edu.cn;
chongxuanli@ruc.edu.cn; dcszj@tsinghua.edu.cn

## Abstract

Diffusion models have obtained substantial progress in image-to-video generation. However, in this paper, we find that these models tend to generate videos with less motion than expected. We attribute this to the issue called conditional image leakage, where the image-to-video diffusion models (I2V-DMs) tend to over-rely on the conditional image at large time steps. We further address this challenge from both inference and training aspects. First, we propose to start the generation process from an earlier time step to avoid the unreliable large-time steps of I2V-DMs, as well as an initial noise distribution with optimal analytic expressions (Analytic-Init) by minimizing the KL divergence between it and the actual marginal distribution to bridge the training-inference gap. Second, we design a time-dependent noise distribution (TimeNoise) for the conditional image during training, applying higher noise levels at larger time steps to disrupt it and reduce the model's dependency on it. We validate these general strategies on various I2V-DMs on our collected open-domain image benchmark and the UCF101 dataset. Extensive results show that our methods outperform baselines by producing higher motion scores with lower errors while maintaining image alignment and temporal consistency, thereby yielding superior overall performance and enabling more accurate motion control. The project page: `https://cond-image-leak.github.io/`.

## 1 Introduction

Image-to-video (I2V) generation aims to generate videos with dynamic and natural motion while maintaining the content of the given image. It allows users to guide video creation from the input image (and optional text), thus increasing controllability and flexibility in content creation. Like the remarkable progress in text-to-image (T2I) generation [42, 39, 21, 8, 18, 5] and text-to-video (T2V) generation [10, 46, 6, 11], diffusion models have also obtained promising results for I2V generation [9, 15, 63, 12, 69, 70]. However, such models are not fully understood.

In this paper, we observe that existing image-to-video diffusion models (I2V-DMs) tend to generate videos with less motion than expected (see Fig. 1). We attribute this to a previously overlooked issue called *conditional image leakage* (see Sec. 3.1). Normally, the noisy input contains the motion information of the target video, and I2V-DMs should rely on it to predict motion, while the static conditional image provides content guidance. However, in practice, as the diffusion process

---

*Equal contribution.    †Correspondence to: C. Li and J. Zhu.

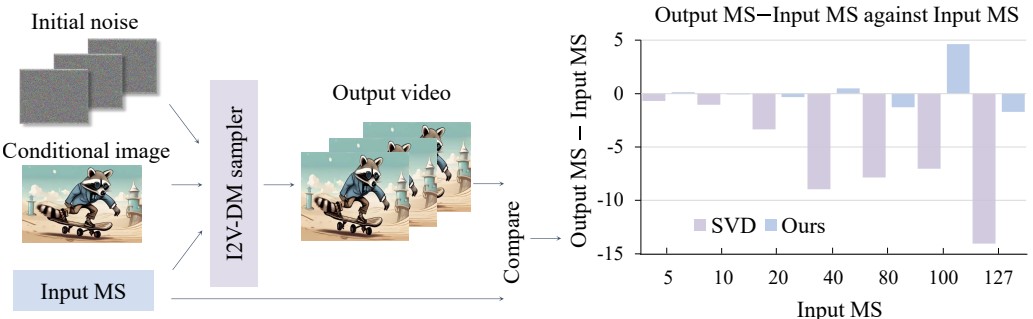

Figure 1: **The issue of existing I2V-DMs.** Regardless of input motion scores (Input MS), the output motion scores (Output MS) are consistently lower than expected. In contrast, our method yields output motion scores either higher or lower than Input MS with reduced error.

progresses—especially at large time steps, the noisy input becomes heavily corrupted, while the conditional image preserves extensive detail of the target video. This biases the model to over-rely on the conditional image and neglect the noisy input, leading to videos with reduced motion. To validate this, we corrupt the ground truth (GT) clean video and compute one-step clean video predictions at each time step. As shown in Fig. 2, the predicted clean videos exhibit markedly reduced motion than GT at large time steps, indicating leakage.

Based on the above analysis, we attempt to address the challenge from both inference and training aspects (see Sec. 3.2). First, we present a simple yet effective inference strategy that starts the video generation process from an earlier time step, thus avoiding the unreliable large-time steps of the I2V-DMs. To further enhance performance, we derive an initial noise distribution with optimal analytic expressions (Analytic-Init) by minimizing the KL divergence between it and the true marginal distribution to bridge the training-inference gap. Second, to mitigate leakage during training, we propose a time-dependent noise distribution (TimeNoise) that increases noise levels at larger time steps, effectively disrupting the conditional image and reducing model dependency on it. This is achieved by employing a logit-normal distribution with a center that gradually shifts over time. Finally, our method achieves higher motion scores with reduced motion score error and ensures that the predicted clean video maintains motion dynamics comparable to the ground truth across all time steps, effectively mitigating conditional image leakage. Notably, our general strategies are adaptable to various I2V-DMs based on both VP-SDE [12, 63, 70] and VE-SDE [9] framework.

Empirically, we validate our methods on various I2V-DMs [9, 63, 12, 70] using our collected open-domain images (ImageBench) and the UCF101 dataset. We conduct a user study on ImageBench and report FVD [55], IS [7], and motion score [9] on UCF101. For motion-conditioned models [9, 70], we also report the motion score error between the generated video and the input motion score at different levels. Extensive experimental results demonstrate that our strategies outperform baselines by producing higher motion scores with lower errors while maintaining image alignment and temporal consistency, thereby yielding superior overall performance and enabling more accurate motion control.

## 2 Background

**Diffusion Models.** Diffusion models gradually perturb the data $\boldsymbol{x}_0 \sim q(\boldsymbol{x}_0)$ via a forward diffusion process and reverse the process to recover it. The forward transitional kernel $q_{t|0}(\boldsymbol{x}_t|\boldsymbol{x}_0)$ is given by

$$\boldsymbol{x}_t = \alpha_t \boldsymbol{x}_0 + \sigma_t \boldsymbol{\epsilon}, \quad \boldsymbol{\epsilon} \sim \mathcal{N}(\mathbf{0}, \boldsymbol{I}), \quad t \in [0, T], \tag{1}$$

where $\alpha_t$ and $\sigma_t$ are the noise schedule chosen to ensure that $\boldsymbol{x}_T$ contains minimal information about $\boldsymbol{x}_0$. Such forward diffusion processes can be viewed as stochastic differential equations (SDEs), among which two prevalent types are commonly used [48, 29]. One is the variance-preserving SDE (VP-SDE) [42, 24], where $\alpha_t^2 + \sigma_t^2 = 1$ with $\alpha_t \to 0$ as $t \to T$, ensuring $p_T(\boldsymbol{x}_T) = \mathcal{N}(\boldsymbol{x}_T; \mathbf{0}, \boldsymbol{I})$. The other is the variance exploding SDE (VE-SDE), where $\alpha_t \equiv 1$ and $\sigma_T$ is set to a large constant, resulting in $p_T(\boldsymbol{x}_T) \approx \mathcal{N}(\boldsymbol{x}_T; \mathbf{0}, \sigma_T^2 \boldsymbol{I})$. Such models can be parameterized with a noise-prediction model $\epsilon_\theta(\boldsymbol{x}_t, t)$ ($\epsilon$-prediction) [24], and the parameters are learned by minimizing:

$$\mathbb{E}_{\boldsymbol{x}_0 \sim q(\boldsymbol{x}_0), \boldsymbol{\epsilon} \sim \mathcal{N}(\mathbf{0}, \boldsymbol{I}), t \sim \mathcal{U}(1, T)} \left[ \|\boldsymbol{\epsilon}_\theta(\boldsymbol{x}_t, t) - \boldsymbol{\epsilon}\|_2^2 \right], \tag{2}$$

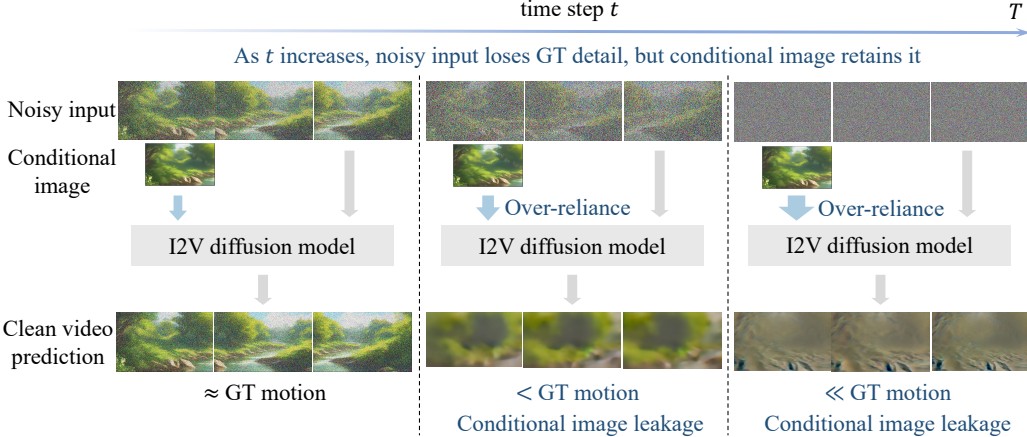

Figure 2: **Identifying conditional image leakage**. As time step progresses, the noisy input becomes heavily corrupted, whereas the conditional image retains considerable detail from GT. This biases the model to over-rely on the conditional image at large $t$, resulting in videos with less motion than GT.

where the noisy input $x_t \sim q_{t|0}(x_t|x_0)$. Alternative parametrizations such as $x_0$-prediction [29], $v$-prediction [45], $F$-prediction [28] are also commonly applied. The $\epsilon$-prediction and $x_0$-prediction aims to predict the added noise or clean video from noisy input $x_t$. Starting from $x_T \sim p_T(x_T)$, various samplers [47, 33, 34, 28] can be employed to generate data. More recent studies [66] further demonstrate that diffusion models can generate realistic images with controllable semantics given only a few labels.

**Diffusion Models for Image-to-Video Generation.** Given an image $y_0$ from the open domain, the goal of I2V is to generate a video $X_0 = \{x_0^i\}_{i=1}^N$ with dynamic and natural motion while keeping alignment with the appearance of $y_0$. This task can be formulated as designing a conditional distribution $p_\theta(X_0|y_0)$, which is achieved by a conditional diffusion model minimizing:

$$\mathbb{E}_{X_0, y_0, \epsilon, t} \left[ \|\epsilon_\theta(X_t, y_0, t) - \epsilon\|_2^2 \right], \tag{3}$$

where $X_t \sim q_{t|0}(X_t|X_0)$. Typically, $y_0$ is the first frame of $X_0$ and DynamiCrafter [63] adopts a randomly selected frame from $X_0$ as $y_0$. The key issue is to effectively integrate the conditional image $y_0$ into the diffusion model. Most methods use CLIP image embeddings [40] to maintain the semantic content of $y_0$. Notably, VideoCrafter1 [12] and Dynamicrafter [63] employ the last layer's full patch visual tokens from the CLIP ViT, enriching the encoded information, and other approaches prefer the class token layer. Yet, solely depending on these embeddings, such as in VideoCrafter1 [12], compromises detail retention, resulting in degradation of the image alignment. To enhance detail representation, I2VGen-XL [69] combines the conditional image with the noisy initial frame, while VideoComposer [57] develops a STC-encoder for multiple conditions. Although superior to CLIP image embeddings, these strategies still fail to fully retain the conditional image content. To mitigate this, AnimateAnything [15], Dynamicrafter [63] and SVD [9] directly concatenate noisy video $X_t$ with $y_0$, which injects detailed information to the model. Apart from the prior work mentioned before, we discuss other related work about diffusion models for image generation and video generation in the Appendix E.

## 3 Method

Although existing I2V-DMs discussed in Sec. 2 have achieved significant progress, such models are not fully understood. In this section, we first identify a critical yet previously overlooked issue in I2V-DMs: conditional image leakage (CIL) (see Sec. 3.1). We then address this issue from both inference and training aspects accordingly (see Sec. 3.2). Finally, we offer insights into existing I2V-DMs through the lens of CIL (see Sec. 3.3).

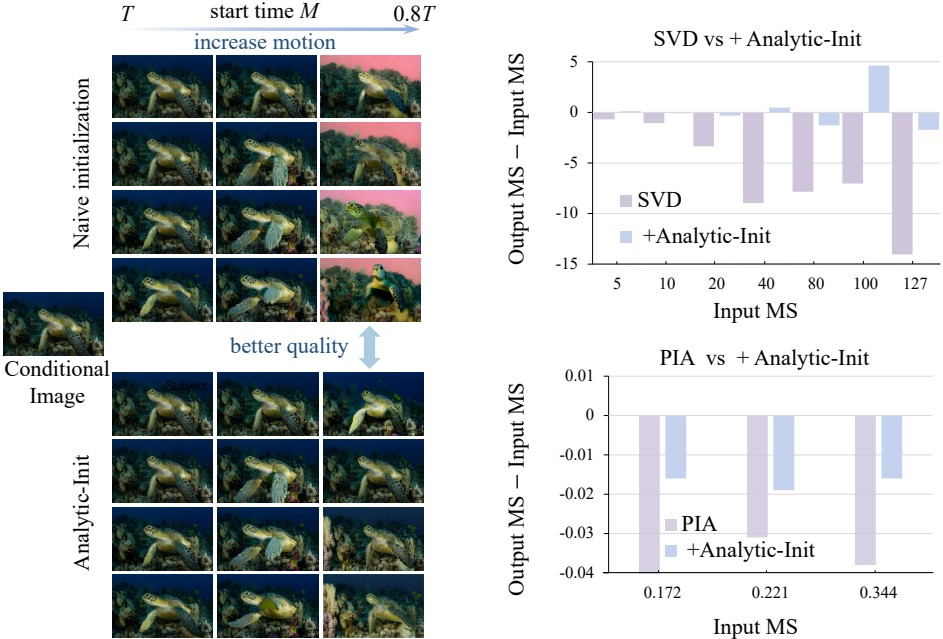

(a) Impact of tuning $M$ and Analytic-Init on visual quality.   (b) Impact of Analytic-Init on the motion score.

Figure 3: **Benefits of Analytic-Init.** (a) An early start time $M$ enhances motion but a too-small $M$ degrades visual quality due to the training-inference gap, which Analytic-Init helps to reduce. (b) Analytic-Init produces higher motion scores with lower errors, mitigating conditional image leakage.

## 3.1 Identifying Conditional Image Leakage in Image-to-video Diffusion Models

As shown in Fig. 1, we observe that, regardless of the input motion scores, the motion scores of generated videos from existing I2V-DMs [9, 70] are consistently lower than expected. This raises the question: why do these models always produce lower motion scores, rather than fluctuating above or below the expected values, as observed in our method?

To understand this, we need to consider the source of motion information in the generated videos, which comes from the noisy input $X_t$. Ideally, I2V-DMs should rely primarily on $X_t$ for motion, with the static conditional image $y_0$ providing content guidance. However, as shown in Fig. 2, at large time steps, the noisy input $X_t$ becomes increasingly corrupted, while the conditional image $y_0$ retains significant information of the target video. This biases the model to over-rely on the conditional image and neglect the noisy inputs, leading to videos with reduced motion.

To validate this, we corrupt a ground truth (GT) clean video $X_0$ via the forward transition kernel in Eq. (1) and use it as the noisy input to compute the one-step prediction $\hat{X}_{t \to 0}$ at time $t$:

$$\hat{X}_{t \to 0} = (X_t - \sigma_t \boldsymbol{\epsilon}_\theta(X_t, y, t))/\alpha_t. \tag{4}$$

Ideally, $\hat{X}_{t \to 0}$ should predict the GT $X_0$ from noisy input $X_t$ and exhibit comparable motion dynamics. However, as shown in Fig. 2, as time progresses—particularly at large time steps, $\hat{X}_{t \to 0}$ exhibits markedly reduced motion than GT, indicating the conditional image leakage. This results in videos with reduced motion starting from time $T$. Notably, recent techniques that adjust the noise schedule towards higher noise levels [14, 27, 9, 32] may further exacerbate this issue (see Appendix C for details).

## 3.2 Solving Conditional Image Leakage in Image-to-video Diffusion Models

Building upon the above analysis, this section presents general strategies to address the issue of conditional image leakage in both the inference and training aspects.

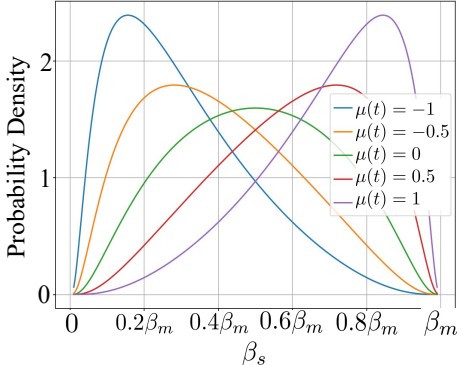
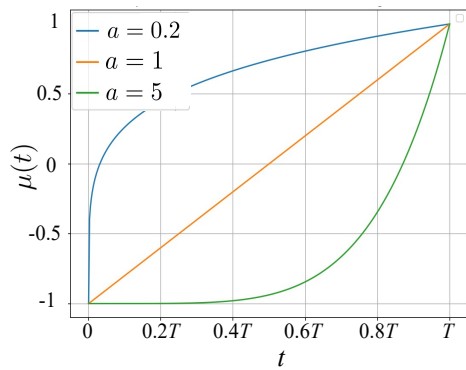

(a) TimeNoise $p_t(\beta_s)$ with hyperparameters $\beta_m$ (maximum noise) and $\mu(t)$ (distribution center).

(b) Distribution center $\mu(t) = 2t^a - 1$ with $a$ controlling monotonic behavior flexibly.

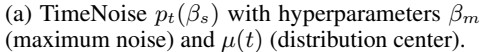
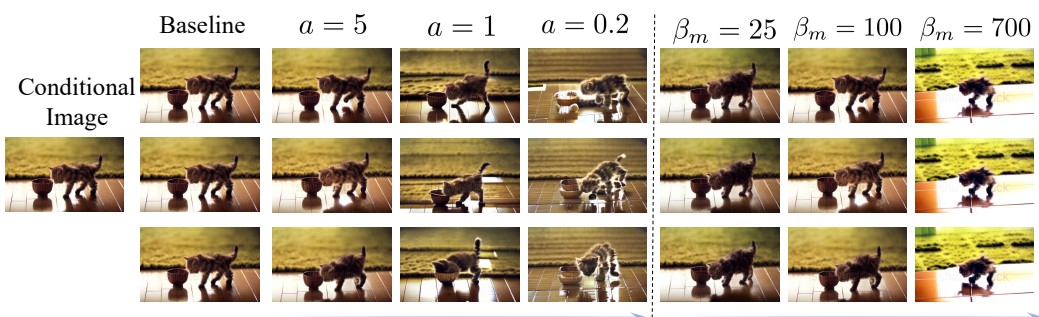

(c) Effects of tuning $a$ in $\mu(t)$ and $\beta_m$, with higher noise levels enhancing motion but reducing alignment.

Figure 4: **Visualization of TimeNoise and the impact of tuning its hyperparameters.** (a) The designed $p_t(\beta_s)$ favors high noise levels at large $t$, gradually shifting to lower noise levels as $t$ decreases. This is achieved by (b) $\mu(t)$ increasing monotonically with $t$. Finally, (c) modifying $a$ and $\beta_m$ enables a trade-off between dynamic motion and image alignment.

**Inference strategy.** As discussed in Sec. 3.1, conditional image leakage easily occurs at large time steps. To this end, a straightforward solution is to start the generation process from an earlier time step $M \in (0, T)$, thus avoiding the unreliable later stages of I2V-DMs. Let $p_M(X_M)$ denote the initial noise distribution at the start time $M$. Initially, we set $p_M(X_M) = p_T(X_T)$, i.e. $\mathcal{N}(\mathbf{0}, \mathbf{I})$ in VP-SDE [24, 63, 12, 70] or $\mathcal{N}(\mathbf{0}, \sigma_T^2 \mathbf{I})$ in VE-SDE [29, 9]. As illustrated in Fig. 3 (a), this straightforward strategy markedly improves motion dynamics without sacrificing other performance. However, a smaller $M$ value (e.g., $M = 0.8T$) results in poor visual quality due to the training-inference discrepancy.

To mitigate this gap, we propose Analytic Noise Initialization (*Analytic-Init*) to refine the initial noise distribution $p_M(X_M)$ by minimizing the KL divergence between it and the true marginal distribution $q_M(X_M)$ of the forward diffusion process. Inspired by previous work [4, 3, 64], we demonstrate that when $p_M(X_M)$ is modeled as a normal distribution $\mathcal{N}(X_M; \boldsymbol{\mu}_p, \sigma_p^2 \mathbf{I})$, the optimal mean $\boldsymbol{\mu}_p^*$ and variance $\sigma_p^{2*}$ have analytical solutions, as stated in Proposition 1.

**Proposition 1.** *Given a normal distribution $p_M(X_M) = \mathcal{N}(X_M; \boldsymbol{\mu}_p, \sigma_p^2 \mathbf{I})$ and $q_M(X_M)$ is the margin distribution of diffusion forward process at time $M$, with the forward trainsition kernel $q_{M|0}(X_M|X_0) = \mathcal{N}(X_M; \alpha_M X_0, \sigma_M^2 \mathbf{I})$, the minimization problem $\min_{\boldsymbol{\mu}_p, \sigma_p^2} D_{KL}(q_M(X_M) || p_M(X_M))$ yields the following optimal solution:*

$$\boldsymbol{\mu}_p^* = \alpha_M \mathbb{E}_{q(X_0)}[X_0], \quad \sigma_p^{2*} = \alpha_M^2 \frac{\sum_{j=1}^{d}[Var(X_0^{(j)})]}{d} + \sigma_M^2, \tag{5}$$

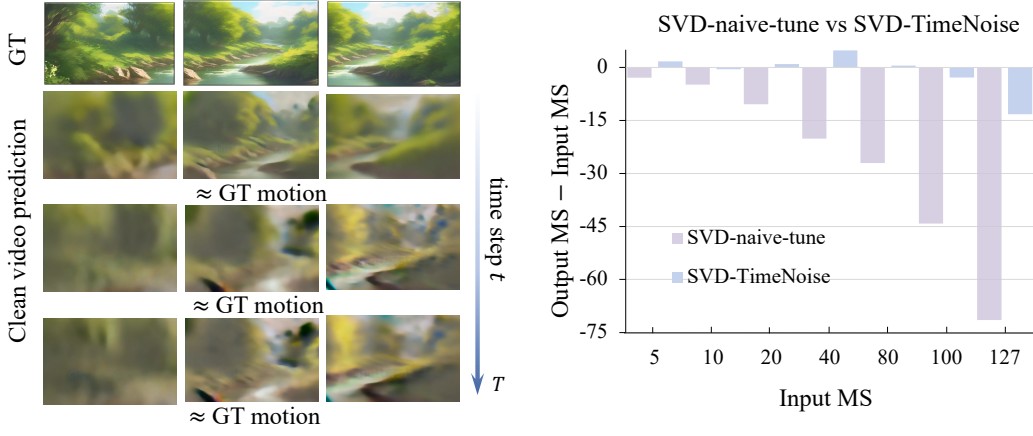

(a) One-step clean video prediction of TimeNoise.

(b) Impact of TimeNoise on the motion score.

Figure 5: **Benefits of TimeNoise.** TimeNoise (a) generates $\hat{X}_{t \to 0}$ that maintains motion dynamics comparable to the GT across all time steps, and (b) achieves higher motion scores with lower errors, effectively reducing conditional image leakage.

*where $q(X_0)$ denotes the data distribution , $d$ denotes the dimension of the data, and $X_0^{(j)}$ denotes the $j$-th component of $X_0$.*

The proof of Proposition 1 is provided in Appendix A. Empirically, $\boldsymbol{\mu}_p^*$ and $\sigma_p^{2*}$ can be estimated using the method of moments [4]. The steps for inference are outlined in Algorithm 1. As demonstrated by the qualitative results in Fig. 3 (a) and quantitative results in Tab. 8, Analytic-Init improves video quality by reducing the training-inference gap, especially for smaller $M$. Finally, as shown in Fig. 3 (b) and Tab. 1, Analytic-Init produces higher motion scores with lower errors, allowing for more accurate motion control and reducing conditional image leakage.

**Training strategy.** In this section, we show how to address the issue of conditional image leakage during the training phase. As outlined in Sec. 3.1, the conditional image $\boldsymbol{y}_0$ retains substantial details of the target video, causing I2V-DMs to rely heavily on it. To mitigate this, a natural approach is to perturb $\boldsymbol{y}_0$ to relieve this dependency. Our first attempt is to introduce noise at a similar scale to that in $X_t$, aiming to balance the model's challenge of predicting clean video from $X_t$ or $\boldsymbol{y}_0$, thereby lessening reliance on $\boldsymbol{y}_0$. However, this strategy also makes it difficult to employ $\boldsymbol{y}_0$, resulting in lower video quality.

To overcome this, we propose a noise distribution on $\boldsymbol{y}_0$ that introduces substantial noise to prevent leakage while maintaining a cleaner $\boldsymbol{y}_0$ to aid content generation. Given that $X_t$ contains less information about $X_0$ as time progresses, increasing the risk of leakage, we further develop a time-dependent noise distribution $p_t(\beta_s)$ (TimeNoise). The key principle is to favor high noise levels at large time steps to sufficiently disrupt $\boldsymbol{y}_0$, shifting towards lower noise levels as the time step decreases. To achieve this, we employ a logit-normal distribution [18, 1] defined as below:

$$p_t(\beta_s; \mu(t), \beta_m) = \frac{\beta_m}{\sqrt{2\pi}} \frac{1}{\beta_s(\beta_m - \beta_s)} e^{-\frac{(\text{logit}(\frac{\beta_s}{\beta_m}) - \mu(t))^2}{2}}, \tag{6}$$

where $\text{logit}(\frac{\beta_s}{\beta_m})$ follows a normal distribution centered around $\mu(t)$ with a standard deviation of 1. This noise distribution includes two hyperparameters: $\beta_m$, the maximum noise level, and $\mu(t)$, the center of the distribution. As illustrated in Fig. 4 (a), we can adjust $\mu(t)$ over time $t$ to satisfy the previously mentioned design principle. Finally, the noisy conditional image $\boldsymbol{y}_s$ at time $t$ is obtained by $\boldsymbol{y}_s = \boldsymbol{y}_0 + \beta_s \boldsymbol{\epsilon}$, where $\beta_s \sim p_t(\beta_s)$, $\boldsymbol{\epsilon} \sim \mathcal{N}(\mathbf{0}, \boldsymbol{I})$. We also tried other adding noise choices but found them to be less effective (see Appendix D). During inference, we add a fixed noise level to the conditional image across all time steps, following CDM[25], as the model is trained with varying noise levels. Empirically, we find directly using the clean conditional image performs well and thus adopt it for simplicity. The full algorithm is shown in Algorithm 2.

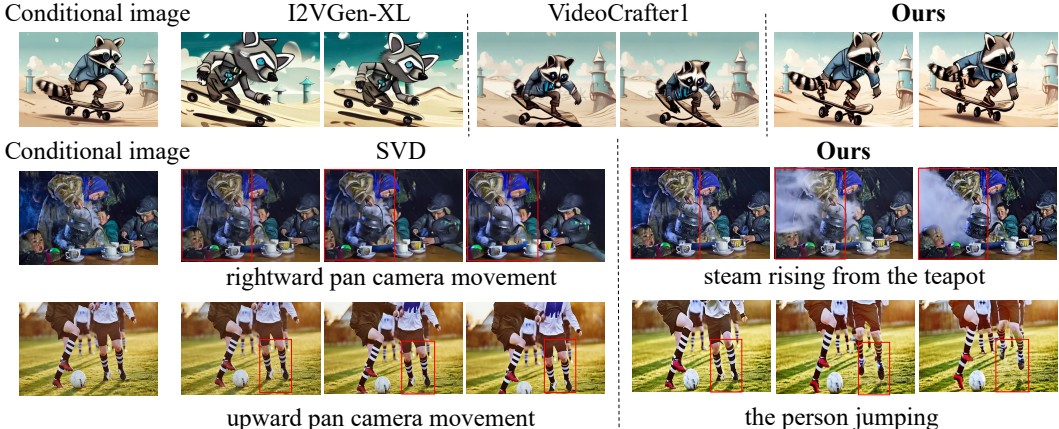

Figure 6: **Understanding exiting work from conditional image leakage.** I2VGen-XL [69] and VideoCrafter1 [12] mitigates the leakage at the expense of image alignment. The SVD produces videos with camera movements while keeping objects relatively static to meet high motion scores, while ours generates videos that feature both natural and dynamic object movements.

Next, we conduct a systematic analysis of the two hyperparameters to investigate their impact on video generation. Firstly, $\mu(t)$ is designed to increase monotonically with time, ranging from $\mu(0) = -1$ to $\mu(T) = 1$. To formalize this, we define $\mu(t)$ as a power function: $\mu(t) = 2t^a - 1$, where $a > 0$. This formulation allows flexible tuning of $a$ to control the monotonic behavior, where smaller values of $a$ cause higher noise levels to be sampled at later time steps. As shown in Fig. 4 (b), (1) $a = 1$ corresponds to a linear increase; (2) $a \in (0, 1)$ represents a concave function, indicating a faster noise level increase; and (3) $a > 1$ corresponds to a convex function, indicating a slower increase in noise levels over time. As shown in Fig. 4 (c), higher noise levels (e.g., when $a < 1$) lead to increased dynamic motion but reduced temporal consistency and image alignment. For the maximum noise level $\beta_m$, the only constraint is that it must be greater than 0. As shown in Fig. 4 (c), a higher $\beta_m$ enhances dynamic motion but decreases temporal consistency and image alignment, while a lower $\beta_m$ reduces motion. Additionally, we apply TimeNoise to the CLIP Image Embedding for both VideoCrafter1 [12] and DynamiCrafter [63], as they use full patch visual tokens from CLIP, which contain substantial information about the conditional image, increasing the likelihood of leakage.

Finally, we replicate the experiments described in Sec. 3.1, with results presented in Fig. 5 and Tab. 1. These results demonstrate that TimeNoise achieves higher motion scores with reduced error and ensures that $\hat{X}_{t\to 0}$ maintains motion dynamics comparable to the ground truth across all time steps, effectively mitigating conditional image leakage.

### 3.3 Understanding Existing Work from Conditional Image Leakage

In this section, we analyze popular I2V-DMs [63, 9, 70, 12, 69] through the lens of CIL. Although these models do not explicitly address this issue, we believe their strategies mitigate it to some degree.

Firstly, some methods only use partial information from the conditional image, which can help reduce leakage. For example, VideoCrafter1 [12] only utilizes CLIP Image Embedding, and I2VGEN-XL [69] adds $y_0$ to the first frame of the noisy input. However, as shown in Fig. 6, the videos generated by these methods do not fully capture the details of $y_0$. To address this, models like Dynamicrafter [63] directly incorporate $y_0$ into I2V-DMs, improving detail preservation but also increasing the risk of leakage. Dynamicrafter selectively refines spatial layers while preserving the pre-trained temporal layers, which contain motion priors and thus maintain motion dynamics to a certain extent. However, it does not inherently solve the leakage issue. Moreover, some methods introduce external signals [9, 70, 65, 62, 60], forcing the model to align with additional conditions, which reduces its dependence on $y_0$ and helps mitigate leakage. However, we argue that they do not address the core challenge of I2V-DMs, which should predict clean video primarily from noisy input to capture motion information. As illustrated in Fig. 6, the SVD often results in static objects with excessive camera movements to meet high motion score requirements. In contrast, our method generates videos with natural, vivid object movements. In summary, while the above methods mitigate

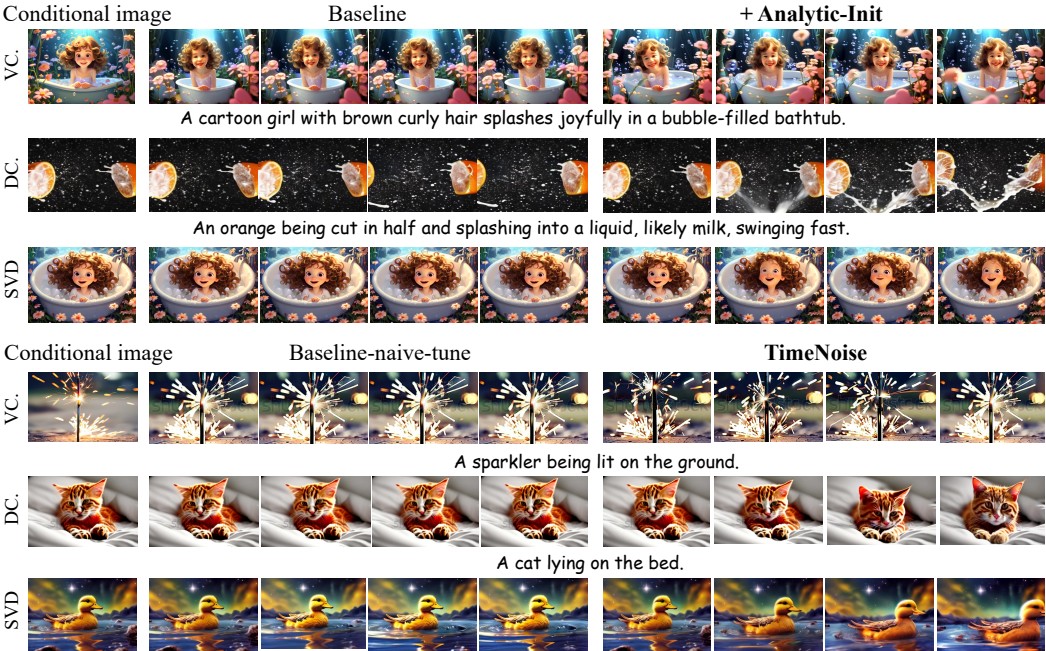

Figure 7: **Qualitative results of TimeNoise and Analytic-Init applied to various I2V-DMs.** Ours significantly enhances video dynamism while maintaining image alignment and temporal consistency. VC. and DC. denote VideoCrafter1 [12] and DynamiCrafter [63] respectively.

CIL to some extent, our approach provides a more effective solution to the fundamental challenges in I2V-DMs, enabling more precise motion control and enhancing naturalness by focusing more on the noisy input.

## 4 Experiments

### 4.1 Setup

**Datasets.** We use WebVid-2M [2] as the training dataset, with all videos resized and center-cropped to $320 \times 512$ at 16 frames and 3 fps. For evaluation, we use UCF101 [49] and our ImageBench dataset, which includes diverse categories (e.g., nature, humans, animals, plants, food, vehicles) and complex elements like numerals, colors, and intricate scenes, similar to DrawBench [44]. In total, 100 images are collected from various websites and T2I models such as SDXL [39] and UniDiffuser [5].

**Evaluation Metrics.** On UCF101, we report Fréchet Video Distance (FVD), Inception Score (IS), and Motion Score (MS). For ImageBench, we conduct user studies with 10 subjects to perform pairwise comparisons of our methods against baselines, evaluating motion, temporal consistency, image alignment, and overall performance. For motion-conditioned methods, we also report the motion score error between the generated video and the input motion score at various levels. Motion scores are computed using flow maps following SVD [9], except for the PIA [70], where we follow its original algorithm and compute the $L1$ distance. More details on the metric computations are provided in Appendix B.

**Implementation Details.** For Analytic-Init, we use 5000 samples from the Webvid-2M dataset to estimate $p_M(X_M)$ by default. We set $M = 0.96T, 0.96T$ and $\sigma_M = 100$ for VideoCrafter1 [12], DynamiCrafter [63], and SVD [9] on UCF101. For ImageBench, we adjust $M$ to $0.92T, 0.92T$ and $\sigma_M = 100$. For the TimeNoise, we set $\beta_m = 25, 100$, and 100 for VideoCrafter1, DynamiCrafter, and SVD, respectively. The function $\mu(t) = 2t^5 - 1$ is applied across all baselines. These baselines are fine-tuned for 20,000 iterations, using either the official code or replication of the official settings except for batch size. More detailed information can be found in the Appendix B.

Table 1: **Quantitative results of TimeNoise and Analytic-Init applied to various I2V-DMs on the UCF101 dataset.** <Method>-naive-tune represents a naively fine-tuned baseline using the same training setup as our TimeNoise, ensuring a fair comparison. <Method>-CIL denotes the full version using both TimeNoise and Analytic-Init.

| Model | FVD↓ | IS↑ | Motion Score ↑ |
|---|---|---|---|
| DymiCrafter [13] | 363.8 | 16.39 | 50.96 |
| DymiCrafter + Analytic-Init | **316.3** | **17.66** | **71.04** |
| DymiCrafter-naive-tune | 382.5 | 21.12 | 31.68 |
| DymiCrafter-naive-tune + Analytic-Init | **342.9** | **22.71** | **50.08** |
| DymiCrafter-TimeNoise | **334.9** | 21.42 | 72.32 |
| DymiCrafter-CIL | **332.1** | **22.84** | **73.92** |
| VideoCrafter1 [12] | 353.9 | 18.75 | 63.36 |
| VideoCrafter1 + Analytic-Init | **341.6** | **19.86** | **139.04** |
| VideoCrafter1-naive-tune | 460.3 | 23.98 | 62.72 |
| VideoCrafter1-naive-tune + Analytic-Init | **450.1** | **24.50** | **65.12** |
| VideoCrafter1-TimeNoise | **452.2** | **24.62** | **64.80** |
| VideoCrafter1-CIL | **443.7** | **25.11** | **66.7** |
| SVD [9] | 388.3 | 36.32 | 16.64 |
| SVD + Analytic-Init | **382.0** | **36.81** | **19.68** |
| SVD-naive-tune | 311.0 | 22.03 | 9.60 |
| SVD-naive-tune + Analytic-Init | **277.1** | **22.18** | **20.64** |
| SVD-TimeNoise | **272.2** | **23.01** | **20.96** |
| SVD-CIL | **272.4** | **25.18** | **21.44** |

## 4.2 Results

**The effectiveness of our inference and training strategy.** We validate our strategies on following I2V-DMs: VideoCrafter1 [12], DynamiCrafter [63] and SVD [54]. We validate our inference strategy against the original method and a naively finetuned version, and our training strategy against a naively finetuned version. The quantitative comparisons and qualitative results are presented in Tab. 1, Fig. 3, Fig. 5 and Fig. 7, leading to several key observations. *First*, our strategies significantly improve motion scores and reduce motion score error, demonstrating their effectiveness in precisely controlling motion degrees and mitigating conditional image leakage. *Second*, we achieve both dynamic motion and high video quality, as reflected by improved FVD and IS scores. The user study in Fig. 8 further supports that our strategies enhance video dynamism while preserving image alignment and temporal consistency, yielding superior overall performance.

**Evaluating the combined inference and training strategies.** In this section, we aim to investigate the necessity of combining two strategies. As illustrated in Tab. 7, for DynamiCrafter [63] and SVD [9], our TimeNoise effectively mitigates conditional image leakage, rendering the additional inference strategy less impactful. Conversely, for VideoCrafter1 [12], which relies solely on CLIP image embedding for information, employing excessive TimeNoise disrupts image alignment. Hence, we utilize a moderate TimeNoise ($\beta_m = 25$), where the inference strategy remains effective.

**Comparison with SOTA noise initialization and conditioning augmentation methods.** We compare our Analytic-Init with SOTA noise initialization methods [41, 61, 19] and our TimeNoise with SOTA conditioning augmentation [25]. As shown in Tab. 2, Analytic-Init outperforms all baselines. It achieves higher motion scores than FrameInit [41], which limits motion by duplicating the conditional image as static guidance. Compared to FreeInit [61], our method reduces inference time by more than half while slightly improving performance, as FreeInit's iterative process increases time. Compared to Progressive Noise [19], Analytic-Init performs better by generating videos from an earlier time step, a gap the baseline cannot handle effectively. As shown in Fig. 9 in the Appendix, CDM [25] results in lower motion, whereas our method produces more dynamic videos. Additionally, a naive baseline using a single value for $\beta_s = \beta_m \frac{\mu(t)+1}{2}$ leads to poor image alignment, further demonstrating the effectiveness of our time-dependent noise distribution.

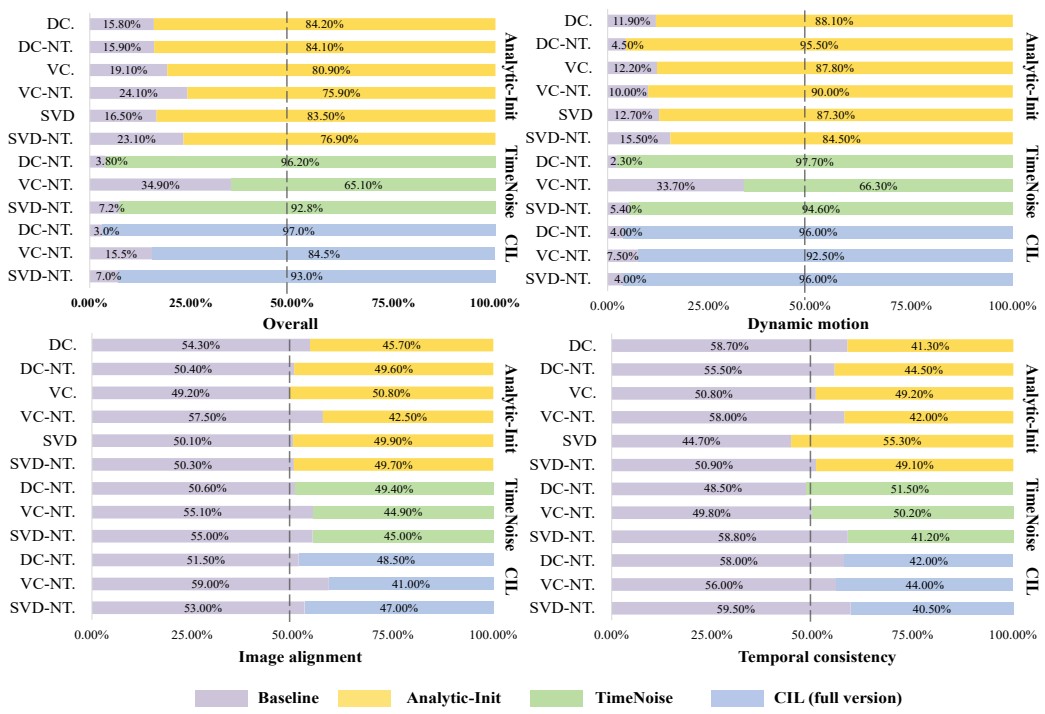

Figure 8: **User preference percentages for ours against baselines**. <Method>-NT. is the naively fine-tuned baseline. DC. and VC. denote DynamiCrafter and VideoCrafter1. Our strategies significantly enhance video dynamism while maintaining image alignment and temporal consistency, thus achieving superior results overall.

Table 2: **Comparison with SOTA noise initialization methods.** Rate. is the average user ranking of each method based on overall performance. Time measures the duration needed to generate a video.

| Method | FVD↓ | IS↑ | MS↑ | Rate.↓ | Time↓ |
|---|---|---|---|---|---|
| FrameInit [41] | 380.7 | 20.09 | 32.16 | 4.57 | 24.3s/it |
| FreeInit [61] | 347.4 | 22.66 | 46.24 | 1.95 | 49.6s/it |
| Progressive Noise [19] | 358.1 | 21.35 | 49.76 | 3.31 | 23.3s/it |
| Analytic-Init | **342.9** | **22.71** | **50.08** | **1.77** | **22.6s/it** |

## 5 Conclusions and Discussions

In this paper, we identify a common issue in I2V-DMs: conditional image leakage. We address this challenge from two aspects. First, we introduce an inference strategy that starts the generation process from an earlier time step to avoid the unreliable late-time steps of I2V-DMs. Second, we design a time-dependent noise distribution for the conditional image to mitigate conditional image leakage during training. We validate the effectiveness of these strategies across various I2V-DMs.

**Limitations and broader impact.** One limitation of this paper is the need to balance TimeNoise to prevent conditional image leakage while maintaining image integrity. While we demonstrate the effectiveness of our strategy on existing I2V-DMs, we do not provide a definitive choice for a scratch-trained model. We leave this in the future work. Furthermore, we must use the method responsibly to prevent any negative social impacts, such as the creation of misleading fake videos.

## Acknowledgement

This work was supported by NSFC Projects (Nos. 62350080, 62106122, 92248303, 92370124, 62350080, 62276149, U2341228, 62076147), Beijing NSF (No. L247030), Beijing Nova Program (No. 20230484416), Tsinghua Institute for Guo Qiang, and the High Performance Computing Center, Tsinghua University. J. Zhu was also supported by the XPlorer Prize.

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

# A Proof of Proposition 1

According to Lemma 2 of Analytic-DPM [4], the KL divergence between initial noise distribution $p_M(X_M)$ and the actual marginal distribution $q_M(X_M)$ can be expressed as:

$$D_{KL}(q_M(X_M)||p_M(X_M)) = D_{KL}(\mathcal{N}(X_M; \boldsymbol{\mu}_q, \Sigma_q)||\mathcal{N}(X_M; \mu_p, \sigma_p^2 I))$$
$$+ H(\mathcal{N}(X_M; \boldsymbol{\mu}_q, \Sigma_q)) - H(q_M(X_M)), \quad (7)$$

where $\boldsymbol{\mu}_q, \Sigma_q$ denote expectation and covariance matrix of $q_M(X_M)$, and $H(\cdot)$ denotes the entropy of a distribution. Since $q_M(X_M)$ is determined by the forward diffusion process and is independent of the $\mu_p$ and $\sigma_p^2$ in $p_M(X_M)$, the last two terms of Eq. (7) can be considered as constants. Therefore, the optimization objective can be formulated as:

$$\min_{\mu_p, \sigma_p^2} D_{KL}(q_M(X_M)||p_M(X_M)) \Leftrightarrow \min_{\mu_p, \sigma_p^2} D_{KL}(\mathcal{N}(X_M; \mu_q, \Sigma_q)||\mathcal{N}(X_M; \mu_p, \sigma_p^2 I)). \quad (8)$$

According to the property of the KL divergence between two normal distributions, $D_{KL}(\mathcal{N}(X_M; \boldsymbol{\mu}_q, \Sigma_q)||\mathcal{N}(X_M; \mu_p, \sigma_p^2 I))$ could be expanded as:

$$\frac{1}{2}\left[\frac{1}{\sigma_p^2}\|\boldsymbol{\mu}_p - \boldsymbol{\mu}_q\|^2 + d\log(\sigma_p^2) + \frac{1}{\sigma_p^2}tr(\Sigma_q) - \log\det(\Sigma_q) - d\right], \quad (9)$$

where $d$ denotes the dimension of the data. Minimizing Eq. (9) yields:

$$\boldsymbol{\mu}_p^* = \boldsymbol{\mu}_q = \mathbb{E}_{q_M(X_M)}[X_M]. \quad (10)$$

Taking the derivative of Eq. 9 with respect to $\sigma_p^2$, the optimal $\sigma_p^2$ comes to:

$$\sigma_p^{2*} = \frac{tr(Cov_{q_M(X_M)}[X_M]) + \|\boldsymbol{\mu}_p - \boldsymbol{\mu}_q\|^2}{d}, \quad (11)$$

$$= \frac{tr(Cov_{q_M(X_M)}[X_M])}{d}, \quad (12)$$

$$= \frac{\sum_{j=1}^{d}(Var(X_M^{(j)}))}{d}. \quad (13)$$

We now represent $\boldsymbol{\mu}_p^*$ and $\sigma_p^{2*}$ with $X_0$. Taking the $X_M$ defined in Eq. (1) into Eq. (10), the optimal $\boldsymbol{\mu}_p^*$ could be further represented as

$$\mu_p^* = \alpha_M \mathbb{E}_{q(X_0)}[X_0] + \mathbb{E}_{q(\epsilon)}[\boldsymbol{\epsilon}], \quad (14)$$

$$= \alpha_M \mathbb{E}_{q(X_0)}[X_0]. \quad (15)$$

Similarly, for the optimal variance $\sigma_p^{2*}$, we can derive

$$Var(X_M^{(j)}) = Var(\alpha_M X_0^{(j)} + \sigma_M \boldsymbol{\epsilon}^{(j)}) \quad (16)$$

Given that $X_0$ and $\epsilon$ are independent of each other, the variance $Var(X_M^{(j)})$ can be further decomposed as:

$$Var(X_M^{(j)}) = Var(\alpha_M X_0^{(j)}) + Var(\sigma_M \boldsymbol{\epsilon}^{(j)}), \quad (17)$$

$$= \alpha_M^2 Var(X_0^{(j)}) + \sigma_M^2. \quad (18)$$

Finally, taking Eq. (18) into Eq. (13), the optimal $\sigma_p^{2*}$ can be represented as

$$\sigma_p^{2*} = \alpha_M^2 \frac{\sum_{j=1}^{d}[Var(X_0^{(j)})]}{d} + \sigma_M^2. \quad (19)$$

# B Implementation Details

## B.1 Code Used and License

We validate our strategies on DynamiCrafter [63] ($320 \times 512$ version), VideoCrafter1 [12] ($320 \times 512$ version), and SVD [9]. All used codes in this paper and their licenses are listed in Tab. 3.

Table 3: **Code Links and Licenses.**

| Method | Link | License |
|---|---|---|
| VideoCrafter1 [12] | `https://github.com/AILab-CVC/VideoCrafter` | Apache License |
| DynamiCrafter [63] | `https://github.com/Doubiiu/DynamiCrafter` | Apache License |
| SVD | `https://github.com/Stability-AI/generative-models` | MIT License |
| UniDiffusers [5] | `https://github.com/thu-ml/unidiffuser` | AGPL-3.0 license |
| SDXL | `https://github.com/Stability-AI/generative-models` | Open RAIL++-M |

Table 4: **Training settings for Dynami-Crafter [63] and VideoCrafter1 [12].**

| Config | Value |
|---|---|
| Optimizer | AdamW |
| Learning rate | 1e-5 |
| Weight decay | 1e-2 |
| Optimizer momentum | $\beta_1, \beta_2$=0.9, 0.999 |
| Batch size | 64 |
| Training iterations | 20,000 |

Table 5: **Training settings for SVD [9].**

| Config | Value |
|---|---|
| Optimizer | AdamW |
| Learning rate | 3e-5 |
| Weight decay | 1e-2 |
| Optimizer momentum | $\beta_1, \beta_2$=0.9,0.999 |
| Batch size | 48 |
| Training iterations | 20,000 |

Table 6: **Compute resources**.

| Model | Iterations | GPU-type | GPU-nums | Hours |
|---|---|---|---|---|
| DynamiCrafter [63] | 20,000 | A800 | 8 | 8 |
| VideoCrafter1 [12] | 20,000 | A800 | 8 | 8 |
| SVD [9] | 20,000 | A800 | 6 | 7 |

## B.2 Training and Inference Details

We utilize the official training code of DynamiCrafter (refer to Tab. 3) to fine-tune both Dynami-Crafter [63] and VideoCrafter1 [12], and reproduce the training code for SVD by ourselves. Throughout the training phase, we maintained consistent settings across all models, with the sole exception of incorporating our TimeNoise component for a fair comparison. Each model was fine-tuned using the WebVid-2M dataset [2], where videos were resized and center-cropped to dimensions of $320 \times 512$ and segmented into sequences of 16 frames. In light of the discussions in Sec. 3.3 regarding the adverse effects of motion scores and the lack of a precise method by SVD [9] to compute these scores, we set a fixed motion score of 20 during training. We fix the frame rate at 3 fps and use dynamic frame rates for DynamiCrafter [63] and VideoCrafter1 [12]. Additional details can be found in Tab. 4 and Tab. 5. Our experiments were conducted using A800-80G GPUs, and the computational costs are detailed in Tab. 6. For inference, the official model codes were used for sampling (see Tab. 3). Specifically, we employed a DDIM sampler with 50 steps for DynamiCrafter [63] and VideoCrafter1 [12], and Heun's 2nd order method with 25 steps for SVD [9].

## B.3 Evaluation

**FVD and IS.** Following prior studies [26, 10], we compute the Fréchet Video Distance (FVD) and Inception Score (IS) for 2,048 and 10,000 samples on the UCF101 dataset, respectively. Specifically, the FVD is calculated using the code available at `https://github.com/SongweiGe/TATS/` with a pre-trained I3D action classification model, which can be accessed at `https://www.dropbox.com/scl/fi/c5nfs6c422nlpj880jbmh/i3d_torchscript.pt?rlkey=x5xcjsrz0818i4qxyoglp5bb8&dl=1`. The IS is derived using the code from `https://github.com/pfnet-research/tgan2`, employing a pre-trained C3D model [51]. For this process, we sample 16 frames at 3 fps, resize them to the default resolution of each model, and use the first frame as the conditional image to generate videos. The FVD and IS are then computed between the generated videos and the ground truth videos. For DynamiCrafter [63] and VideoCrafter1 [12], which utilize text as an additional condition, we employ categories as the textual input.

**Motion Score.** The Motion Scores of DynamiCrafter [63], VideoCrafter1 [12] and SVD are implemented following SVD [9].Specifically, we compute the motion score by resizing the video to $800 \times 450$, extracting dense optical flow maps using RAFT [50] between adjacent frames, calculating the magnitude of the 2D vector for each pixel, averaging these magnitudes spatially, and then summing them across all frames.

As for PIA [70], we calculate the motion score by making a slight modification to the affinity score proposed in [70], namely: motion score = 1 - affinity score.

More specifically, for each video, we calculate the L1 distance between each frame $v^i$ and the condition frame $v^1$ in HSV space, which is denoted as $d^i$. Then we apply this operation to all frames of video clips in the dataset and find the maximum distance value $d_{max}$. We normalize the distance $d^i$ to $[0, 1]$ via $d_{max}$. Finally, the motion score for each frame can be calculated by $m^i = d^i/d_{max} \times (m_{max} - m_{min})$, where $m_{max}$ and $m_{min}$ are hyperparameters set as $1, 0.2$ respectively.

**User study.** We ask users to compare ours with the baselines and determine which ones exhibit more dynamic and natural motion, greater temporal consistency, better alignment with the conditional image, and overall preference.

Table 7: **Evaluating the combined inference (Inference.) and training (Train.) strategies.** Rat. denotes a user study that ranks methods. Refer to Sec. 4.2 for detailed analysis.

| Inference. | Train. | VideoCrafter1 [12] | | | DynamiCrafter [63] | | | SVD [9] | | |
|---|---|---|---|---|---|---|---|---|---|---|
| | | FVD↓ | IS↑ | Rat.↓ | FVD↓ | IS↑ | Rat.↓ | FVD↓ | IS↑ | Rat.↓ |
| ✗ | ✗ | 460.3 | 23.98 | 4 | 382.5 | 21.12 | 4 | 311.0 | 22.03 | 4 |
| ✓ | ✗ | 450.1 | 24.50 | 1.9 | 342.9 | 22.71 | 2.9 | 277.1 | 22.18 | 2.9 |
| ✗ | ✓ | 452.2 | 24.62 | 2.7 | 334.9 | 21.42 | **1.4** | **272.2** | 23.01 | 1.6 |
| ✓ | ✓ | **443.7** | **25.11** | **1.4** | **332.1** | **22.84** | 1.7 | 272.4 | **25.18** | 1.5 |

## C  The Influence of Adjusting the Noise Schedule

In this section, we first examine commonly used strategies that involve adjusting the noise schedule to higher noise levels to bridge the training-inference gap [14, 27, 9, 32], which, unfortunately, may exacerbate conditional image leakage. Following the SVD [9], we increase noise by adjusting the $P_{mean}$ in the EDM framework [28]. Surprisingly, as shown in Fig. 10, we observe that larger $P_{mean}$ values correspond to reduced motion in the generated videos. We hypothesize that this is due to the increased noise added to $X_t$ by larger $P_{mean}$, making it more challenging to predict clean frames and thus more prone to conditional image leakage. Consequently, the synthesized videos tend to lack dynamic and vivid motion.

## D  Ablation Studies for Adding Noise Method

In this section, we show another choice to add noise to the conditional image. Specifically, the noisy conditional image $y_s$ at time $t$ is obtained by

$$y_s = (1 - \beta_s)y_0 + \beta_s\epsilon, \tag{20}$$

where $\beta_s \sim p_t(\beta_s), \epsilon \sim \mathcal{N}(\mathbf{0}, \mathbf{I})$, and $\beta_m = 1$. As shown in Fig. 12, this choice leads to video discoloration in SVD [9].

## E  More Related Work on Diffusion Models for Image and Video Generation

**Diffusion Models for Image Generation.**  Recently, diffusion models have achieved significant breakthroughs in image, video, and 3D generation [24, 16, 42, 38, 5, 71, 37, 58, 59, 72]. For image generation, the latent diffusion model [42] addresses computational costs by leveraging VQ-VAE [53] to transform pixel-space images into compact latent representations, subsequently training diffusion models within this latent space. Building upon the latent space concept, subsequent works such as

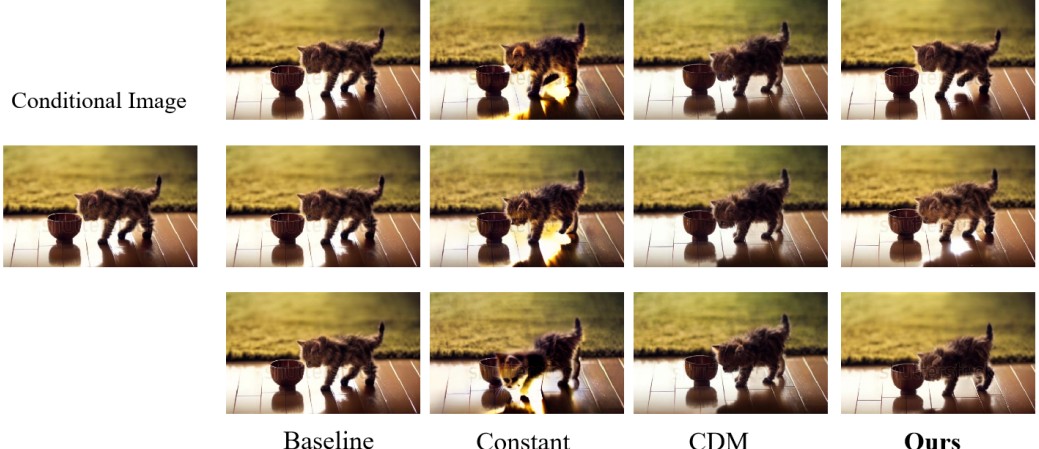

Conditional Image

Baseline      Constant      CDM      **Ours**

Figure 9: **The qualitative comparison between our TimeNoise and baselines mentioned in Sec. 3.2.** The constant results in poor image alignment, while the CDM [25] shows low motion. Ours achieves the best visual quality.

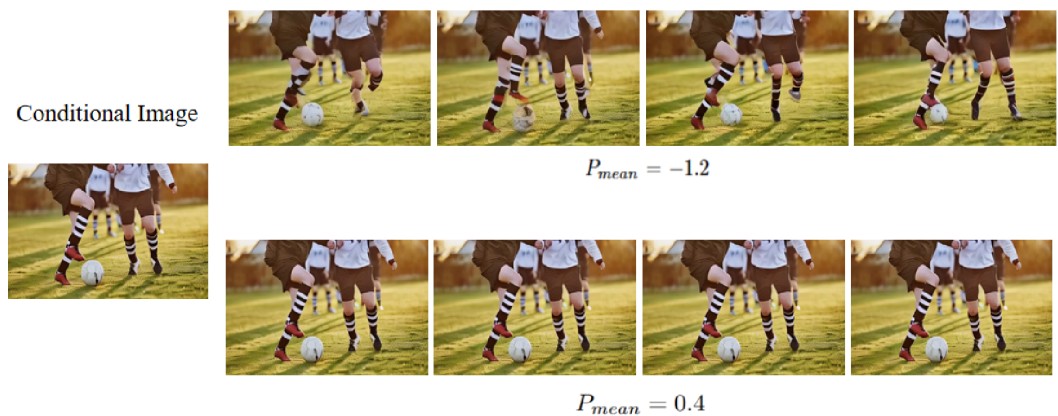

Figure 10: **Adjusting the noise schedule towards more noise further exacerbate conditional image leakage.**

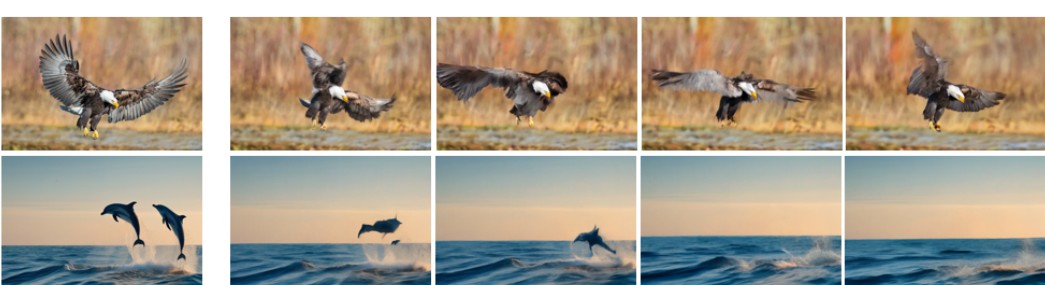

Figure 11: An inappropriate motion score leads to poor temporal consistency.

Conditional Image

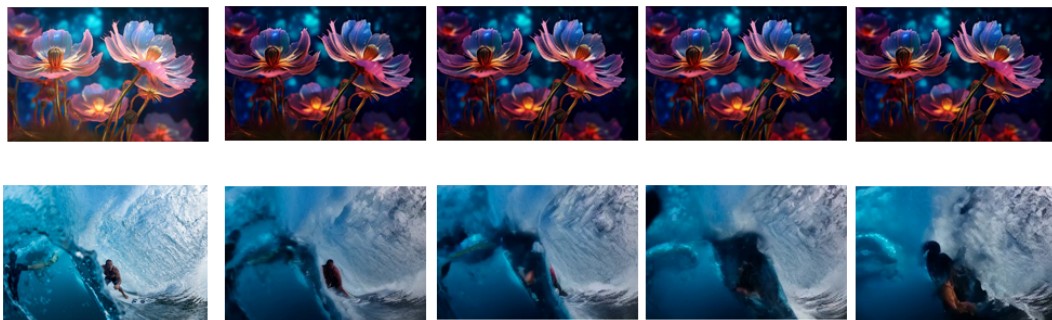

Figure 12: **Other choice of adding noise on conditional image.** Adding noise via Eq. 20 causes video discoloration in SVD [9].

Table 8: Effects of tuning $M$ and Analytic-Init.

| Model | $M$ | FVD↓ | IS↑ |
|---|---|---|---|
| | T | 363.8 | 16.39 |
| | 0.98T | 345.1 | 16.57 |
| | 0.96T | 343.1 | 17.54 |
| DynamiCrafter [63] | 0.94T | **325.2** | 19.12 |
| | 0.92T | 365.8 | 20.49 |
| | 0.9T | 442.2 | **22.81** |
| | 0.98T | 324.3 | 16.44 |
| | 0.96T | **316.3** | 17.66 |
| DynamiCrafter + Analytic-Init | 0.94T | 319.6 | 19.13 |
| | 0.92T | 347.2 | 20.58 |
| | 0.9T | 378.6 | **22.40** |

Table 9: Quantitative results of the inference strategy across varying initial time $M$ on the UCF101 dataset.

| Model | $M$ | FVD↓ | IS↑ |
|---|---|---|---|
| DynamiCrafter [63] | T | 363.8 | 16.39 |
| | 0.96T | **316.3** | 17.66 |
| | 0.92T | 347.2 | 20.58 |
| + Analytic-Init | 0.88T | 407.2 | 23.32 |
| | 0.84T | 535.1 | **24.27** |
| | 0.8T | 696.1 | 23.67 |
| VideoCrafter1 [12] | T | 353.9 | 18.75 |
| | 0.96T | **341.6** | 19.86 |
| | 0.92T | 344.3 | 21.58 |
| + Analytic-Init | 0.88T | 368.4 | 21.82 |
| | 0.84T | 400.8 | **21.90** |
| | 0.8T | 445.6 | 21.21 |

| Model | $\sigma_M$ | FVD↓ | IS↑ |
|---|---|---|---|
| SVD-finetune [9] | 700 | 311.0 | 22.03 |
| | 500 | 301.9 | 22.00 |
| | 300 | 290.2 | 21.94 |
| + Analytic-Init | 100 | 277.1 | 22.18 |
| | 70 | **272.5** | **22.27** |
| | 50 | 295.5 | 21.89 |

**Algorithm 1** Sampling from an I2V diffusion model with Analytic-Init

---

**Require:** the conditional image $y_0$, the sampler for diffusion models $Sampler(\cdot, \cdot, \cdot)$
    select the initial time step $M \in (0, T)$
    calculate $\boldsymbol{\mu}_p^*$ and $\sigma_p^{2*}$ in initial noise distribution according to Eq. (5)
    $X_M \sim \mathcal{N}(X_M; \boldsymbol{\mu}_p^*, \sigma_p^{2*}\boldsymbol{I})$
    **for** $t = M, ..., 1$ **do**
        $X_{t-1} = Sampler(X_t, y_0, t)$
    **end for**
    **return** $X_0$

---

**Algorithm 2** Training an I2V diffusion model with TimeNoise

---

**Require:** the clean video $X_0$, the conditional image $y_0$, the noise schedule $\alpha_t, \sigma_t$, the time-dependent
    noise distribution $p_t(\beta_s)$
    **repeat**
        $t \sim \mathcal{U}([0, T])$
        $\beta_s \sim p_t(\beta_s)$                            ▶ Sampling the noise level for the conditional image
        $\boldsymbol{\epsilon}_y \sim \mathcal{N}(\boldsymbol{0}, \boldsymbol{I})$
        $\boldsymbol{y}_s = \boldsymbol{y}_0 + \beta_s \boldsymbol{\epsilon}_y$                    ▶ Adding noise to the conditional image
        $\boldsymbol{\epsilon} \sim \mathcal{N}(0, \boldsymbol{I})$
        $X_t = \alpha_t X_0 + \sigma_t \boldsymbol{\epsilon}$
        $\theta \leftarrow \theta - \eta \nabla_\theta \|\boldsymbol{\epsilon}_\theta(X_t, \boldsymbol{y}_s, t) - \boldsymbol{\epsilon}\|_2^2$
    **until** converged

---

SDXL [39], DALLE-3 [8], and SD3 [18] have further enhanced the performance. By exploiting such advancements in text-to-image diffusion models, numerous methods have demonstrated promising results in text-driven controllable image generation and image editing [68, 22, 52]. As for image editing, studies such as Prompt-to-Prompt [22] and Plug-and-Play [52] have explored attention-based control mechanisms over generated content, consistently delivering impressive results. For image translation, EGSDE [71] and DiffuseIT [30] propose to employ an additional energy function to guide the inference process of a pre-trained diffusion model.

**Diffusion Models for T2V Generation.** Approaches to T2V generation can be classified into two main categories. The first one involves directly generating videos based on text [46, 10, 23, 12, 13, 54, 67, 56, 9, 17]. For instance, as a pioneering endeavor, Make-A-Video [46] utilizes a pre-trained text-to-image model along with a prior network for T2V diffusion models, obviating the necessity for paired video-text datasets. VideoLDM [10] maintains the parameters of a pre-trained T2I model while fine-tuning the additionally introduced temporal layers. While many models are based on a U-Net [43] architecture, more recently, transformers have emerged as a foundational architecture for video generation due to their scalability [21, 36, 35]. The second category typically entails a two-step generation process: first generating an image based on the textual input and subsequently creating a video conditioned on the text and the generated image [31, 20]. For instance, Emu-video [20] initializes a factorized text-to-video model using a pre-trained text-to-image model and then fine-tunes temporal modules in the I2V stage.

