# OpenReview forum: "Identifying and Solving Conditional Image Leakage in Image-to-Video Diffusion Model"
_NeurIPS.cc/2024/Conference — NeurIPS 2024 poster_

### Official Review · Reviewer_f9AW · 2024-06-27

**Soundness:** 3
**Presentation:** 2
**Contribution:** 2
**Rating:** 5
**Confidence:** 5

**Summary:**

This paper argue that I2V-DMs tend to overly rely on the conditional image at large time steps, resulting in videos with less dynamic motion.
To address this, they propose using the KL divergence between the initial noise distribution and the actual marginal distribution to enhance the amplitude of motion while preserving video quality.
Besides，the paper propose a time-dependent noise distribution for the conditional image during training process.

**Strengths:**

Motivation. The challenge of conditional image leakage presented in the paper is a credible issue and represents a significant academic problem.

The paper is exceptionally well-organized, making complex concepts easily digestible for the reader.

**Weaknesses:**

1. Claim. Regarding the conditional image and the diversity of content changes (referred to as 'motion' in this paper), it is, in fact, a matter of trade-off. The essence of image to video generation is to leverage the conditional image to enhance video quality. Therefore, the Proposition 1 and Training Strategy proposed by the authors are essentially fine-tuning techniques that favor the diversity of motion.

2. Additionally, the foundation model of video generation guided only by the first frame inherently contains uncertainty in its output; it does not know whether the user wants to generate a video with significant or minimal motion. Some recent studies (last frame guided[1], motion control[2])have demonstrated that incorporating additional guidance signals can indeed achieve big and precise motion.

Particularly, the works DragNUWA and MotionCtrl have shown that for the image-to-video generation task, it is still possible to generate significant motion. Therefore, the foundation model of SVD has the potential to directly generate substantial motion. In this context, the initial version of SVD, which could only generate minor motion, might be due to the lack of a specific condition or guidance signal for significant motion, rather than an inherent incapability. From a subjective standpoint, not all applications require substantial motion.

[1] ToonCrafter: Generative Cartoon Interpolation
[2] MotionCtrl: A Unified and Flexible Motion Controller for Video Generation
[3] DragAnything: Motion Control for Anything using Entity Representation
[4] DragNUWA: Fine-grained Control in Video Generation by Integrating Text, Image, and Trajectory

3. Experiments. The experiments appear to be somewhat confusing and not entirely convincing.
Regarding the variables of the related training strategy displayed in Figure 8, the reviewer has not found the corresponding ablation study, which is a core contribution of this paper. Additionally, the author has not provided an ablation study on the 'M' of Proposition 1, such as testing performance changes between 0.6T and 1T, which is central to the first contribution. If the reviewer has overlooked it, please point it out.

Additionally, Figure 9 also appears to be somewhat confusing. Was a user study not conducted on the Analytic-Init? What does '-tune' signify—does it mean finetuning (training strategy)? If so, why does it appear in the inference phase?

--------------------
After reading the author’s and other reviewers' feedback, I have adjusted my score to 5.

**Questions:**

Can the authors provide more insights into the selection of hyperparameters (Line 230 Implementation Details) for the time-dependent noise distribution?

**Limitations:**

yes, the authors have discussed the limitations.

---

> ### Author Rebuttal · Authors · 2024-08-06
>
> # Author Response to Reviewer f9AW
>
> We thank reviewer f9AW for the valuable and constructive comments. We address the concerns as follows.
>
> ## Q1: Claim. The conditional image and the diversity of content changes are, in fact, a matter of trade-off. The essence of image-to-video generation is to leverage the conditional image to enhance video quality. Proposition 1 and the Training Strategy proposed by the authors are essentially fine-tuning techniques that favor the diversity of motion.
>
> We appreciate the reviewer for the comment but disagree with our highest respect. While the conditional image for video quality and motion diversity can be a trade-off, **we emphasize that our results simultaneously achieve dynamic motion and high video quality, without sacrificing either aspect**. As demonstrated in Table 1, we achieve superior FVD and IS metrics compared to all baselines, indicating higher video quality. The user study in Figure 9 further confirms that our method outperforms all baselines in overall aspects, including video quality and dynamic motion. The results in **Rebuttal Table A** show that our method achieves a lower motion score error than all baselines, underscoring that our method helps to control motion degrees more precisely.
>
> ## Q2. Recent studies show incorporating additional guidance signals can achieve significant motion and control the motion degree. From a subjective standpoint, not all applications require substantial motion.
>
> We clarify that our method is orthogonal to the approaches that introduce additional signals and our method enables them control motion degrees more precisely. We validate this by consistently achieving a lower error between the motion score of the generated video and the given desired motion score, regardless of the input motion score in SVD (see **Rebuttal Table A**). More details are provided in Common Concern 1. DragAnything, DragNUWA, MotionCtrl, and ToonCrafter are novel controllable methods that achieve great performance on controllable video generation. We will include a discussion with these methods in the Related Work section in the final version
>
> ## Q3: Ablation studies.
>
> We did a systematic analysis of each hyperparameter in our proposed strategy in the original paper.
>
> For the start time $M$ in Analytic-Init, the qualitative and quantitative results are reported in Figure 4, Table 7 and Table 8. As discussed in lines 144-146 and lines 159-162 of the original paper, an appropriate $M$ can enhance performance by increasing motion without compromising other performance. A too-small $M$ delivers poor visual quality due to the training-inference gap. Analytic-Init helps to mitigate it, especially with a smaller $M$.
>
> For the two hyperparameters in the training strategy, the qualitative results and corresponding analysis are shown in Figure 5 and described in lines 183-195 in the original paper. Specifically, higher noise levels, such as those produced by a concave function ($a$ < 1), enhance dynamic motion but reduce temporal consistency and image alignment. For the maximum noise level $\beta_m$, a higher $\beta_m$ enhances dynamic motion but decreases temporal consistency and image alignment, while a lower $\beta_m$ leads to less motion.
>
> ## Q4: In Figure 9, was a user study not conducted on the Analytic-Init? What does '-tune' signify—does it mean finetuning (training strategy)? If so, why does it appear in the inference phase?
>
> The inference strategy in Figure 9 is denoted as Analytic-Init. The < method >-tune represents a naively finetuned baseline using the same training setup and dataset as our TimeNoise training strategy, ensuring a fair comparison. We validate our inference strategy on both the original baseline and a naively finetuned baseline. The < method >-tune baseline is crucial because I2V-DMs often use private datasets with varying data filtering, resolutions, and unknown training settings. Controlling these variables ensures a fair comparison. We will rename < method >-tune as < method >-naive-tune and label the inference strategy as Analytic-Init for clarity in the final version.
>
> ## Q5: Insights into the selection of hyperparameters
>
> As discussed in lines 184-185 of the original paper, firstly, the principle of $\mu(t)$ is to increase monotonically with time step. This ensures that higher noise levels are favored at later time steps, which carries a higher risk of leakage. To formalize this, we define $\mu(t)$ as a power function: $\mu(t)=2t^a-1, a>0$. This choice allows us to flexibly tune $a$ to achieve various monotonic behaviors, where smaller values of $a$ indicate that more of the later time steps will sample higher noise levels. Specifically:
>
> - $a=1$ corresponds to a linear increase,
> - $a<1$  indicates a concave function, suggesting a faster increase in noise levels over time, and
> - $a>1$ represents a convex function, indicating a slower increase in noise levels over time.
>
> For the choice of the maximum noise level $\beta_m$, the only principle is that it should be greater than 0. We refer to popular choices of maximum noise levels in VE-SDE[1,2] (e.g. 80-100) and experiment with parameters around these values.
>
> [1] Score-based generative modeling through stochastic differential equations.
>
> [2] Elucidating the Design Space of Diffusion-Based Generative Models.

---

> > ### Comment · Reviewer_f9AW · 2024-08-12
> >
> > Thank you for the author's response, which addressed most of my concerns. I will finalize my score after discussing with other reviewers, and it will likely be between 4 and 5. Thanks.

---

> ### Author Response · Authors · 2024-08-12
> **Author Response to Reviewer f9AW**
>
> Dear Reviewer f9AW,
>
> Thank you for your feedback. We are pleased to hear that our responses have addressed most of your concerns. We believe the additional experiments, analysis, and explanation have significantly improved the quality and clarity of our submission. We hope that you and other reviewers may regard this as a sufficient reason to raise the score.
>
> Best, Authors

---

### Official Review · Reviewer_hjyn · 2024-07-05

**Soundness:** 2
**Presentation:** 3
**Contribution:** 3
**Rating:** 6
**Confidence:** 4

**Summary:**

The paper points out that existing image-to-video diffusion models can lead to videos without significant/desired amount of motion. Some evidence is presented to show that this is due to what the authors call "conditional image leakage" where the model places too much emphasis on the conditional image and all the generated frames look too similar to the conditional image. Two strategies are proposed to overcome this issue. The inference strategy starts the reverse diffusion process at an earlier time and uses improved initial noise distribution parameters using a closed-form expression to match the noise parameters at training time. The training strategy develops a time-dependent noise distribution to have higher noise at larger t to place lesser emphasis on the conditioning image for larger t values. Experiments indicate improved video generation using either or both of these strategies.

**Strengths:**

1. The methods presented in the paper are well-motivated and make intuitive sense (although there appear to be some technical issues as noted in the weaknesses below).

2. Results: The quantitative results show clear improvements over baseline. The video examples submitted also show clear improvements over previous methods

3. Relationship to existing works: the authors have done a good job presenting related works carefully and how existing methods may also inherently overcome some issues, if not all.

**Weaknesses:**

1. Issues with derivation: For deriving the expression of better noise initialization, the authors present their proof in Appendix A. The second and third terms of (7) are considered constants for the purpose of optimizing the noise distribution. I don't follow this and think there may be a technical error here. It is written that since $q_M(X_M)$ and $p_M(X_M)$ are independent, the entropy $H(\mathcal{N}(X_M; \mathbf{\mu}_q, \Sigma_q))$ and $H(q_M(X_M))$ can be considered constants for optimizing the parameters $\mathbf{\mu}_q$ and $\Sigma_q$. I don't think this is correct as these parameters influence both $H(\mathcal{N}(X_M; \mathbf{\mu}_q, \Sigma_q))$ and $H(q_M(X_M))$. Can the authors please explain this in detail?

2. Quantifying motion: The authors say that with previous works the models could in principle just output the conditioning image several times and still lead to lower error or improved quality metrics like FID and IS. However, they still use the same metrics for their experiments. Perhaps quantifying amount of motion or even better, quantifying realism of motion directly is important.

3. Figure 9 is not clear. What are orange and blue colors indicating. Please explain this figure in detail and how it shows that the proposed method is considered better by the user study. I see very little differences going from <method> to <method>-tune in the figure.

**Questions:**

No additional questions.

**Limitations:**

Yes, authors have addressed some limitations of their method.

---

> ### Author Rebuttal · Authors · 2024-08-06
>
> # Author Response to Reviewer hjyn
>
> We sincerely thank Reviewer hjyn for the constructive and valuable comments. The concerns are addressed as follows.
>
> ## Q1: Issues with derivation.
>
> We clarify that our objective is to optimize $\mu_p$ and $\sigma_p^2$, the parameters of $p_M(X_M)=N(X_M; \mu_p, \sigma_p^2 I)$, rather than $\mu_q$ and $\Sigma_q$ (as stated in Proposition 1, line 155 of the original paper). Note that $\mu_q$ and $\Sigma_q$ represent the expectation and covariance matrix of the forward marginal distribution $q_M(X_M)$, which is determined by the forward diffusion process and is independent of the $\mu_p$ and $\sigma_p^2$ in $p_M(X_M)$. Given this independence, the second and third terms in Equation (7) can be disregarded for the purpose of optimization. Therefore, the optimization objective can be formulated as:
> $$
>     \min_{\mu_p,\sigma_p^2} D_{KL}(q_M(X_M)||p_M(X_M)) \Leftrightarrow \min_{\mu_p,\sigma_p^2} D_{KL}(N(X_M; \mu_q, \Sigma_q)||N(X_M; \mu_p, \sigma_p^2 I))
> $$
> We will improve the writing of Appendix A in the final version.
>
> ## Q2: The authors say that with previous works the models could in principle just output the conditioning image several times and still lead to lower error or improved quality metrics like FID and IS. However, they still use the same metrics for their experiments. Perhaps quantifying amount of motion or even better, quantifying realism of motion directly is important.
>
> We clarify that models that repeatedly output the conditioning image might achieve a lower **training loss**, as defined by Eq. (2) in the original paper, rather than a lower evaluation error or improved video quality metrics like FVD and IS. In fact, as demonstrated in Table 1 of the original paper, such repetitive behavior can lead to poorer FVD and IS, since these metrics can evaluate the dynamic qualities of videos and static conditioning images lack the inherent dynamics. It's important to note that FVD and IS are video evaluation metrics based on an action classification model rather than image-based metrics like FID and IS, which can reflect video dynamics and video quality.
>
> As suggested, we also incorporate the motion score metric to quantify the amount of motion, which is computed by obtaining dense optical flow maps with RAFT [1] between adjacent frames, calculating the magnitude of the 2D vector for each pixel, and summing these magnitudes across the frame. As demonstrated in **Rebuttal Table F**, our approach yields substantially higher motion scores compared to all baseline methods. We will add this in the final version. Quantifying the realism of motion is indeed crucial, yet, to the best of our knowledge, there currently exists no specific automatic quantitative metric to evaluate this aspect directly. Our user study, which covers overall aspects, takes this into account. We welcome any suggestions from the reviewers regarding additional metrics for evaluating motion realism.
>
> [1] Raft: Recurrent all-pairs field transforms for optical flow.
>
> **Rebuttal Table F**. Motion score results on the ImageBench dataset. < Method >-naive-tune represents a naively fine-tuned baseline using the same training setup and dataset as our TimeNoise training strategy, ensuring a fair comparison.
>
> | Model | Motion Score ↑ |
> |-------|-----------------|
> | DymiCrafter | 50.96 |
> | + Analytic-Init | **71.04** |
> |||
> | DymiCrafter-naive-tune | 31.68 |
> | + Analytic-Init | **50.08** |
> | + TimeNoise | **72.32** |
> |||
> | VideoCrafter1 | 63.36 |
> | + Analytic-Init | **139.04** |
> |||
> | VideoCrafter1-naive-tune | 62.72 |
> | + Analytic-Init | **65.12** |
> | + TimeNoise | **64.80** |
> |||
> | SVD | 16.64 |
> | + Analytic-Init | **19.68** |
> |||
> | SVD-naive-tune | 9.60 |
> | + Analytic-Init | **20.64** |
> | + TimeNoise | **20.96** |
>
> ## Q3: Explanation of Figure 9.
>
> Figure 9 shows user preference percentages for our methods versus baselines. The left side displays baselines, while the right side shows our inference (Analytic-Init) and training (TimeNoise) strategies. Orange indicates the preference for our strategies and blue for the baselines. For instance, in the first row at the top left, 84.2% preferred our inference strategy over the original DynamiCrafter's output (15.8%) from an overall perspective. The < method > and < method >-tune denote the original model and naively finetuned versions, serving as baselines. The < method >-tune uses the same training settings and dataset as our TimeNoise strategy, ensuring a fair comparison. This is crucial because I2V-DMs often use private datasets with varying data filtering, resolutions, and unknown training settings. Controlling these variables ensures a fair comparison.
>
> In summary, we validate our inference strategy against the original method and a naively finetuned version, and our training strategy against a naively finetuned version. As discussed in the original paper (lines 238-245), our strategies significantly enhance video dynamism and natural motion, maintaining image alignment and temporal consistency, thus achieving superior results overall. We will rename < method >-tune as < method >-naive-tune for clarity and improve the caption of Figure 9 in the final version.

---

> > ### Comment · Reviewer_hjyn · 2024-08-13
> > **Thank you for your response**
> >
> > I thank the authors for their response.
> >
> > I under the derivation better now, I think misunderstood some parts initially.
> >
> > The additional results quantifying motion could be of value in the final paper, if accepted. And they are appreciated.
> >
> > Overall, the authors have addressed my concerns, and I believe they have addressed most of the other reviewers' concerns.
> >
> > I am raising my score to a 6 based on NeurIPS rating scale.

---

> > > ### Author Response · Authors · 2024-08-13
> > > **Thanks for the feedback**
> > >
> > > Thank you for the appreciation of our response and the update on the score. We highly appreciate it.

---

### Official Review · Reviewer_NwFJ · 2024-07-10

**Soundness:** 3
**Presentation:** 2
**Contribution:** 3
**Rating:** 6
**Confidence:** 5

**Summary:**

This paper investigates and proposes solutions for the problem of “conditional image leakage” in image-to-video generation. The authors claim that existing image-to-video diffusion models over-rely on the conditional image at large diffusion timesteps when the inputs are too noisy, leading to static video output. To address this issue, the authors propose an inference-time generation process that skips those large timesteps. The authors also propose adding noise to the conditional image during training to mitigate conditional image leakage.

**Strengths:**

This paper identifies the problem of conditional image leakage and shows that existing methods suffer from static motion at large diffusion timesteps. The two proposed solutions empirically boost FVD and IS on UCF101 and bring significant improvements to motion magnitude while maintaining a similar image alignment and temporal consistency in the user study. These show that the proposed approach is indeed useful in generating more dynamic videos.

**Weaknesses:**

There are a lot of prior works [a, b, c] that deal with noise initialization in video diffusion models that are highly related to the problem of conditional image leakage and the proposed inference technique. There are, however, no discussions on the differences between these techniques. Notably, FreeInit [a] and FrameInit [c] perverse the low-frequency component of the initial image – which is at odds with the proposed theory. I believe an in-depth discussion and a comparison with these approaches would help to establish the thesis of this paper. Additionally, PIA [d] also allows motion control – comparing the proposed motion-adding techniques with PIA’s motion control would also be helpful.

[a] FreeInit: Bridging Initialization Gap in Video Diffusion Models

[b] Preserve Your Own Correlation: A Noise Prior for Video Diffusion Models

[c] ConsistI2V: Enhancing Visual Consistency for Image-to-Video Generation

[d] PIA: Your Personalized Image Animator via Plug-and-Play Modules in Text-to-Image Models

Secondly, I am not entirely convinced by the authors’ explanation of conditional image leakage. The authors claim that models learn a shortcut by copying the initial image at large diffusion timesteps, leading to static video output. However, models like VideoCrafter and SVD do produce outputs of diverse motion of some degree, as shown in their papers. What are the authors’ explanations for these motions? Where are they coming from? Trying to copy the conditional image when the input is otherwise noise is likely the optimal solution and would be learned – but that is not a sufficient condition for static output.

Thirdly, how are evaluations normalized for motion magnitude? Methods like SVD and PIA have motion control. Since the evaluation is focused on “dynamic motion”, perhaps choosing a different motion parameter for these methods would lead to different results in the user study. How are those parameters chosen? Do baseline SVD, SVD after finetuning, and SVD with the inference trick interpret the same motion scale parameter in the same way? One way to validate this is to plot the input motion scale parameter against the output motion magnitude (e.g., optical flow) for parameter selection.

**Questions:**

What are the motion magnitudes of SVD in Figure 3 and Figure 7?

Adding different noises to the conditioning image at different time steps requires the conditioning image to be embedded multiple times. How does this affect inference time?

**Limitations:**

The authors have adequately addressed the limitations.

---

> ### Author Rebuttal · Authors · 2024-08-07
>
> # Author Response to Reviewer NwFJ
>
> We thank Reviewer NwFJ for the valuable comments.
>
> ## Q1: Add comparisons.
>
> ### Q1-1: Add comparison with related noise initialization methods.
>
> (1) As suggested, we add comparisons with FreeInit[a], Progressive Noise[b] and FrameInit[c]. Our Analytic-Init outperforms all baselines across all metrics, as demonstrated in **Rebuttal Table C**. Specifically:
> -  compared with FrameInit: we achieve much higher motion scores. This is because FrameInit duplicates the conditional image into a static video and uses it as coarse layout guidance,which can limit motion degree.
> - compared with FreeInit: we reduce inference time by more than half, while achieving a slight enhancement in performance. This is because FreeInit's iterative refinement process, involving the sampling of a clean video, adding noise, and regenerating the video, increases its overall inference time.
> - compared with Progressive Noise: we achieve better performance. This is because our inference framework generates videos from an earlier time step, while the baseline can not handle this training-inference gap.
>
> (2) Additionally, we clarify that our initialization method enhances temporal consistency (referred to as 'low frequency' in related work), consistent with findings in prior work ( see 5th vs 9th columns in Figure 4 of the original paper). Our high-motion capability (referred to as 'high frequency' in related work) is improved by starting the generation process at an earlier time step rather than initialization, avoiding the unreliable late-time steps of I2V-DMs.
>
> Thank you for highlighting these important related works. We will incorporate the above discussions in the final version.
>
> **Rebuttal Table C.** Comparison of different noise initialization methods. Output MS is the motion score of the generated videos, computed via an optical flow network. User Rate. indicates the average rank assigned by users across all methods from overall aspects. Inference Time refers to the duration required to generate a video.
> | Method            | FVD ↓   | IS ↑   | Output MS ↑ | User Rate ↑ | Inference Time ↓ |
> |-------------------|---------|--------|-------------|-------------|------------------|
> | FrameInit         | 380.7   | 20.09  | 32.16       | 4.57        | 24.3s/it         |
> | FreeInit          | 347.4   | 22.66  | 46.24       | 1.95        | 49.6s/it         |
> | Progressive Noise | 358.1   | 21.35  | 49.76       | 3.31        | 23.3s/it         |
> | Analytic-Init     | **342.9** | **22.71** | **50.08**   | **1.77**   | **22.6s/it**     |
>
>
>
> ### Q1-2: Add comparison with PIA.
>
> As suggested, we apply our inference strategy to PIA. Our method consistently enhances the motion degree, regardless of the initial motion level in PIA, as demonstrated in **Rebuttal Table D in the reponse PDF**. This improvement suggests the presence of conditional image leakage in PIA. It is important to note that our method is orthogonal to approaches that introduce additional signals, such as PIA and SVD, and enables them control motion more precisely (see Common Concern 1).
>
> ## Q2: The authors claim models learn a shortcut by copying the image, yet they still produce outputs with motion.
>
> We clarify that the static video example in Figure 2 is a schematic representation. We aim to highlight that these models tend to generate videos with **reduced motion** due to over-reliance on the conditional image at large diffusion timesteps (see lines 6, 34, 103 in the original paper). In some cases, this over-reliance might lead to nearly static videos, as exemplified in Figure 2. We repeat the experiments three times using different random seeds and conducte a statistical analysis using a two-sample t-test between the baselines and ours. As demonstrated in **Rebuttal Table E in the response PDF**, our method achieved higher motion scores with a significance level of p < 0.05 compared to all baselines. We will make it clear in the final version.
>
> ## Q3: The principle for motion score chosen. Plot the input motion score against the output motion score.
>
> For the input motion score chosen, we select a range of motion scores between 5 and 200 on the ImageBench. A score of 20 achieves high visual quality, including high temporal consistency and natural motion, and is used as a baseline. In the remaining experiments (Table 1, Figure 7, Figure 9), we directly use this 20 motion score for the baseline SVD, SVD-tune, and SVD with Analytic-Init. As suggested, we report the input motion scale against the output motion scale in **Rebuttal Table A** and **Figure 1 in the response PDF**. The results show that our method enables SVD control motion degrees more precisely by consistently achieving a lower error between the motion score of outputs and input motion score. See more detailed analysis in Common Concern 1.
>
> ## Q4: The motion score in Figure 3 and Figure 7
>
> In Figure 3, we use the default motion score of 127 in SVD, which tends to show camera movement with poor temporal consistency. In Figure 7, as detailed in Q3, we use a motion score of 20 for its high visual quality. As discussed in Common Concern 1, our method consistently achieves a lower motion score error, regardless of the input motion score in SVD.
>
> ## Q5: How does TimeNoise affect inference time?
>
> During inference, we use the conditional image with a fixed noise level across all time steps, similar to CDM[1], which does not affect inference time. This is because, during training, the model learns to handle conditional images with varying noise levels. In our initial experiments, we tested noise levels at 0, 5, and 10, finding that levels 0 and 5 exhibit similar performance, while a level of 10 sacrifices image alignment and temporal consistency. Based on these findings, we directly use the clean conditional image across all time steps for simplicity. We will include these details in the final version.
>
> [1] Cascaded diffusion models for high-fidelity image generation.

---

> > ### Comment · Reviewer_NwFJ · 2024-08-12
> >
> > I thank the authors for the response.
> > Follow-up questions:
> > - Where is "output motion score" defined? I cannot find it in the rebuttal and the mentions of "motion score" in the main paper seem to be all related to the input, not the output.
> > - Q3 -- The authors chose a score of 20, much lower than the default 127. The authors claim that this "achieves high visual quality". Does this suggest a score of 127 does not achieve high visual quality? Can we back this up? If so, why not 15, or 25? Some tuning has been done for the proposed method (e.g., Table 8) so it would be fair to perform similar tuning for SVD and the baselines.
> > - Also Q3 -- "consistently achieving a lower error between the motion score of outputs and input motion score" -- suggests that the input motion score is supposed to be calibrated to the output motion score. Again I am not sure how the output motion score is defined.
> > -- Comparisons with PIA -- Again, this does not show how the input motion score for PIA is selected.

---

> > > ### Author Response · Authors · 2024-08-13
> > > **Further Response to Reviewer NwFJ**
> > >
> > > We sincerely appreciate the reviewer’s feedback. Below, we address the further concerns in detail.
> > >
> > > ## Q1: Where is "output motion score" defined?
> > >
> > > Sorry for the unclear description. Since the official code is not available, we implement the output motion score following the guidelines for calculating the input motion score as detailed in the Optical Flow section (Appendix C) of the SVD paper. This approach ensures consistency in the calculation of both the input and output motion scores. Consequently, this consistency enables the error to accurately reflect the precision of motion control achieved.
> > >
> > > ## Q2: Some tuning has been done for the proposed method  so it would be fair to perform similar tuning for SVD and the baselines.
> > >
> > > As presented in Rebuttal Table A, we conduct tuning for the input motion score of the two baselines (SVD and SVD-naive-tune) within the range of 5 to 200, including the default score of 127. The results indicate that our method consistently outperforms these baselines, achieving both higher user preference and lower motion score error, regardless of the input motion score.
> > >
> > > ## Q3: A lower error suggests that the input motion score is supposed to be calibrated to the output motion score. I am not sure how the output motion score is defined. Comparisons with PIA -- Again, this does not show how the input motion score for PIA is selected.
> > >
> > > As addressed in our response to Q1, we ensure consistency between the input and output motion scores by implementing the output motion score calculation following the guidelines from the SVD paper. Regarding PIA, since it categorizes motion degree into three discrete levels (small/moderate/large) labeled as 0/1/2, it's not feasible to compute a precise motion score error. Nevertheless, as presented in Rebuttal Table D of the response PDF, regardless of the input motion level, our method consistently outperforms PIA in user preference metrics. Combined with the main baseline SVD results in Rebuttal Table A, these findings underscore the effectiveness of our approach.
> > >
> > > We sincerely appreciate the reviewer’s constructive suggestions and believe that the additional experiments, analysis, and explanations significantly improve the quality of our submission. We hope that this provides a sufficient reason to consider raising the score.

---

> > > > ### Comment · Reviewer_NwFJ · 2024-08-13
> > > >
> > > > Thank you for the clarification. I think Table A and the analysis of input/output motion scores provide important support for the claims made in the paper. Please include them in the main paper. Particularly, please detail the computation of the motion score, that SVD has been trained with such scores, and why it would make sense to directly compare the input/output motion scores. I understand them now but it was not quite clear.
> > > >
> > > > Comparisons with PIA: "it categorizes motion degree into three discrete levels (small/moderate/large) labeled as 0/1/2" -- this is not true. The motion value from PIA comes from pixel-level L1 distance in the HSV space which obviously has more than three levels.

---

> ### Author Response · Authors · 2024-08-13
> **Further Response to Reviewer NwFJ**
>
> We appreciate the reviewer’s quick feedback. Below, we address the further concerns.
>
> ## Q1: Details about the experiments across motion scores.
>
> ### Q1-1: Please detail the computation of the motion score
> We compute the motion score by resizing the video to $800\times450$, extracting dense optical flow maps using RAFT [1] between adjacent frames, calculating the magnitude of the 2D vector for each pixel, averaging these magnitudes spatially, and then summing them across all frames.
>
>
> [1] Raft: Recurrent all-pairs field transforms for optical flow.
>
> ### Q1-2: Why it would make sense to directly compare the input/output motion scores.
>
> Training SVD involves learning a conditional probability distribution $p(X|y)$, where $y$ represents the motion score condition. During inference, the generated output $X' \sim p(X|y)$ is expected to match the given condition $y$. Therefore, the error between the motion score of generated output $y'$ and the given input score $y$ can reflect the discrepancy between the generated outputs and the target distribution in terms of motion magnitude.
>
> We will include the above details in the main paper in the final version.
>
>
> ## Q2: Comparisons with PIA: "it categorizes motion degree into three discrete levels (small/moderate/large) labeled as 0/1/2" -- this is not true. The motion value from PIA comes from pixel-level L1 distance in the HSV space which obviously has more than three levels.
>
> Thanks for highlighting the misunderstanding about PIA. The default three levels (small / moderate / large) of motion degree in PIA correspond to three predefined input affinity scores $S _ {in} = $ {$s _ {in}^i $}$ _ {i=1}^n$ , which are negatively correlated with the motion score, where $n$ is the frame and $s _ {in}^i\in[0,1]$. Consistent with the experiments from Q1, we evaluate the precision of motion control by computing the error between the input and output affinity scores. We implement the output affinity scores $S _ {out} =$ {$s _ {out}^i$}$ _ {i=1}^n$ following the guidelines for calculating the input affinity scores as described in the PIA paper's Inter-frame Affinity section (3.2). Then error between them is computed as:
> $$
>     error = \sum_{i=1}^n \frac{|s_{in}^i - s_{out}^i|}{n}.
> $$
> As shown in **Rebuttal Table G**, our method consistently outperforms the baseline by achieving a lower affinity score error, regardless of the input affinity score. We will include the above experiments in the main paper in the final version.
>
> **Rebuttal Table G.** The error comparison between PIA and PIA with our Analytic-Init. The three motion degree levels of Input Motion Level correspond to three predefined input affinity scores.
> |Input Motion Level|Error(PIA/Ours)$\downarrow$|
> |--|--|
> |Small Motion|0.309/**0.257**|
> |Moderate Motion|0.230/**0.221**|
> |Large Motion|0.182/**0.124**|

---

> > ### Comment · Reviewer_NwFJ · 2024-08-14
> >
> > Thank you for the responses. They addressed most of my concerns. I have increased the rating to weak accept.

---

> ### Author Response · Authors · 2024-08-14
> **Thanks for the feedback**
>
> Thank you for the feedback and for updating the score. We greatly appreciate your patience and the multiple rounds of communication, as your engagement has significantly enhanced the quality of our work.

---

### Official Review · Reviewer_iEBW · 2024-07-14

**Soundness:** 3
**Presentation:** 3
**Contribution:** 3
**Rating:** 7
**Confidence:** 4

**Summary:**

This paper presents an approach for image-to-video (I2V) generation. The authors start with a conditional image leakage problem in existing works where the conditional image significantly influences the generation process thereby impacting the dynamism of the video. The authors propose an inference time and a training time strategy to mitigate the problem. The approach is evaluated on UCF101 and the authors' ImageBench dataset. The results are impressive.

**Strengths:**

1. The authors identified a crucial problem of conditional image leakage with I2V generation.

2. The authors proposed two strategies to address conditional leakage. An inference time strategy that can be readily applied to existing approaches. Another approach is applied during the training time.

3. The paper is well-organized and easy to follow.

4. Authors conducted a user study for qualitative evaluation. The results are impressive.

**Weaknesses:**

1. The authors are encouraged to provide more details about the ImageBench dataset.

2. While performing the user study, what qualities are evaluated by the users?

3. How is the accuracy of the generated videos estimated corresponding to the text prompt?  Do the users rate the matching score of the generated video corresponding to the text prompt?

4. The realism of the generated videos is lacking. This is common for diffusion-based generated visual content.  Have the authors considered how to improve the realism of generated videos?

**Questions:**

Please address the comments in the weaknesses section.

**Limitations:**

Yes

---

> ### Author Rebuttal · Authors · 2024-08-06
>
> # Author Response to Reviewer iEBW
>
> We sincerely thank Reviewer iEBW for the recognition of our work and for providing constructive comments.
>
> ## Q1: Details about the ImageBench dataset.
>
> Our ImageBench dataset is designed based on two key aspects: breadth of categories and logical complexity. For breadth, we include popular categories such as nature, humans (both single and multi-person), animals, plants, buildings, food, arts, and vehicles. For complexity, we incorporate elements like numerals, colors, and complex scenes. Following these principles, we collected 100 images from various sources and T2I models like SDXL[1] and UniDiffuser[2]. We will add this in the revised version.
>
>  [1] Sdxl: Improving latent diffusion models for high-resolution image synthesis.
>
>  [2] One transformer fits all distributions in multi-modal diffusion at scale.
>
> ## Q2: While performing the user study, what qualities are evaluated by the users?
>
> As noted in lines 226-228 of the original paper, users were asked to conduct pairwise comparisons between our method and the baselines, assessing dynamic motion, temporal consistency, image alignment, and overall performance.
>
> ## Q3: Add the evaluation about the text alignment.
>
> We appreciate the constructive feedback provided. Following the suggestion, we add text alignment evaluation using the CLIP-text score[1] and a user study in **Rebuttal Table B**. Since SVD does not utilize text prompts as input, we conducted these evaluations only for VideoCrafter1 and DynamiCrafter. The results demonstrate that our approach achieves comparable text alignment performance with the baselines, suggesting that our strategy does not adversely affect text alignment quality.
>
> [1]: Learning Transferable Visual Models From Natural Language Supervision
>
> **Rebuttal Table B.** Text alignment performance comparison on the ImageBench dataset. < Method >-naive-tune and < Method >-TimeNoise represent finetuned models with naive strategy and our training strategy respectively, using the same setup and dataset for a fair comparison. User preference indicates the percentage of users who preferred our method over the baselines. DC. and VC. denote DynamiCrafter and VideoCrafter1.
>
> | Model | CLIP-text Score ↑ | User Preference ↑ |
> |-------|--------------------|------------------|
> | DC. / DC. + Analytic-Init | 0.274 / **0.277** | **53%** / 47% |
> | DC.-naive-tune / + Analytic-Init | **0.269** / 0.266 | 48% / **52%** |
> | DC.-naive-tune / DC.-TimeNoise | **0.269** / 0.267 | 49% / **51%** |
> ||||
> | VC. / VC. + Analytic-Init | **0.252** / 0.251 | **52%** / 48% |
> | VC.-naive-tune / + Analytic-Init | 0.247 / **0.255** | 45% / **55%** |
> | VC.-naive-tune / VC.-TimeNoise | **0.247** / 0.242 | 46% / **54%** |
>
> ## Q4: Future work on improving the realism of generated videos.
>
> We appreciate the insightful comments. We think there are two promising ways for enhancing the realism of the generated videos. One promising direction is to explore advanced model architectures, such as the Transformer architecture. By refining the architecture to better capture long-range dependencies and spatial-temporal coherence, we may enhance the realism and consistency of the generated videos. Another possible aspect is to scale up the diffusion model. Scaling the model can involve increasing the depth and width of the network, as well as using larger datasets to improve
> quality.

---

### Author Rebuttal · Authors · 2024-08-06

# Common Concerns from reviewers

## Common Concern 1 (from   NwFJ and f9AW): Incorporating additional guidance signals can achieve significant motion and control the motion degree. Show the input motion score against the output motion magnitude.

We clarify that **our method is orthogonal to the approaches that introduce additional signals and enables them to the control motion degree more precisely**. As demonstrated in **Rebuttal Table A** and **Figure 1 in the response PDF**, our method consistently achieves a lower error between the motion score of the generated video and the given desired motion score, regardless of the input motion score. Notably, the baseline tends to generate videos with a significantly lower motion score than the input motion score, suggesting the presence of conditional image leakage. Our method effectively reduces this discrepancy.

In addition, we emphasize that our method not only enables more precise control of the motion degree but also **enhances its naturalness**. As discussed in the original paper (lines 123-130) and illustrated with more examples in **Figure 2 of the response PDF**, the baseline often produces static objects with pronounced camera movements to meet high motion scores. In contrast, our approach generates videos with natural and vivid object movements. This suggests that a simple scalar condition may not be sufficient to address the fundamental challenge facing I2V-DMs, which is to generate clean videos primarily from noisy inputs. Our method offers a more effective solution, enhancing realism and fluidity in the generated videos.



**Rebuttal Table A.** Results on the ImageBench dataset. SVD-naive-tune and SVD-TimeNoise are finetuned SVD with naive finetuning and our TimeNoise respectively, using the same setup for a fair comparison. Input MS is SVD's input motion score. Output MS is the average motion score of generated videos. Error is the absolute difference between them. User Preference is percentages favoring our method. Note that the motion score is implemented based on the description provided in the SVD paper, as the actual code is unavailable. For the first two experiments (SVD-naive-tune and SVD-TimeNoise), both the input and output MS are consistently calculated, ensuring that the error measure is reliable. In the case of the original SVD (the third experiment), the error may not be as robust, but it is included for reference.

| Input MS | Output MS $\uparrow$ | Error $\downarrow$ | User Preference $\uparrow$ |
|----------|-------------|---------|-------------------|
| | **SVD-naive-tune / + Analytic-Init**|
| 5     | 2.08 / **4.64** | 2.92 / **0.36** | 32.0% / **68.0%** |
| 10    | 5.12 / **8.16** | 4.88 / **1.84** | 29.5% / **70.5%** |
| 20    | 9.60 / **20.64** | 10.4 / **0.64** | 16.5% / **83.5%** |
| 40    | 19.84 / **34.08** | 20.16 / **5.92** | 21.0% / **79.0%** |
| 80    | 52.96 / **65.12** | 27.04 / **14.88** | 32.5% / **67.5%** |
| 100   | 55.84 / **83.68** | 44.16 / **16.32** | 19.5% / **80.5%** |
| 127   | 55.52 / **111.20** | 71.48 / **15.80** | 34.0% / **66.0%** |
| 200   | 64.16 / **133.92** | 135.84 / **66.08** | 27.5% / **72.5%** |
| | **SVD-naive-tune / SVD-TimeNoise** |
| 5     | 2.08 / **6.72** | 2.92 / **1.72** | 19.5% / **80.5%** |
| 10    | 5.12 / **9.44** | 4.88 / **0.56** | 9.0% / **91.0%** |
| 20    | 9.60 / **20.96** | 10.4 / **0.96** | 7.2% / **92.8%** |
| 40    | 19.84 / **44.80** | 20.16 / **4.80** | 11.5% / **88.5%** |
| 80    | 52.96 / **80.48** | 27.04 / **0.48** | 15.5% / **84.5%** |
| 100   | 55.84 / **97.12** | 44.16 / **2.88** | 18.5% / **81.5%** |
| 127   | 55.52 / **113.76** | 71.48 / **13.24** | 23.0% / **77.0%** |
| 200   | 64.16 / **150.24** | 135.84 / **49.76** | 35.5% / **64.5%** |
| |**SVD / +Analytic-Init** |
| 5     | 4.32 / **5.12** | 0.68 / **0.12** | 23.0% / **77.0%** |
| 10    | 8.96 / **9.92** | 1.04 / **0.08** | 31.5% / **68.5%** |
| 20    | 16.64 / **19.68** | 3.36 / **0.32** | 16.5% / **83.5%** |
| 40    | 31.04 / **40.48** | 8.96 / **0.48** | 21.0% / **79.0%** |
| 80    | 72.16 / **78.72** | 7.84 / **1.28** | 16.0% / **84.0%** |
| 100   | 92.96 / **104.64** | 7.04 / **4.64** | 38.5% / **61.5%** |
| 127   | 112.96 / **125.28** | 14.04 / **1.72** | 42.0% / **58.0%** |
| 200   | 190.40 / **206.40** | 9.60 / **6.40** | 42.5% / **57.5%** |

---

### Author Response · Authors · 2024-08-12
**Looking forward to further feedback**

Dear AC and Reviewers,

Thank you again for the great efforts and valuable comments. We have carefully addressed the main concerns in detail. We hope you might find the response satisfactory. As the discussion phase is about to close, we are very much looking forward to hearing from you about any further feedback. We will be very happy to clarify any further concerns (if any).

Best, Authors

---

### Decision · Program_Chairs · 2024-09-25

**Decision:**

Accept (poster)

**Comment:**

This paper discusses the issue of conditional image leakage in an image-to-video generation model where the conditioning image may potentially bring unintended control into the generation of video dynamics. The paper proposes two ideas to mitigate the issue, leading to strong empirical performances.

The paper was reviewed by four experts, who questioned various aspects of the paper. Authors provided a strong rebuttal, addressing the concerns, resulting in all reviewers inclining towards acceptance. AC agrees with the reviewers that the paper considers an important problem in conditional video synthesis, and the proposed approach shows solid results. Thus, AC recommends acceptance. Authors should revise the paper as per the reviewers' suggestions as well as incorporate all the comparisons that were provided in the rebuttal in the camera-ready.